# Janus 3D printed dynamic scaffolds for nanovibration-driven bone regeneration

Sandra Camarero-Espinosa [1,2,3] & Lorenzo Moroni [1✉]

The application of physical stimuli to cell cultures has shown potential to modulate multiple cellular functions including migration, differentiation and survival. However, the relevance of these in vitro models to future potential extrapolation in vivo depends on whether stimuli can be applied "externally", without invasive procedures. Here, we report on the fabrication and exploitation of dynamic additive-manufactured Janus scaffolds that are activated on-command via external application of ultrasounds, resulting in a mechanical nanovibration that is transmitted to the surrounding cells. Janus scaffolds were spontaneously formed via phase-segregation of biodegradable polycaprolactone (PCL) and polylactide (PLA) blends during the manufacturing process and behave as ultrasound transducers (acoustic to mechanical) where the PLA and PCL phases represent the active and backing materials, respectively. Remote stimulation of Janus scaffolds led to enhanced cell proliferation, matrix deposition and osteogenic differentiation of seeded human bone marrow derived stromal cells (hBMSCs) via formation and activation of voltage-gated calcium ion channels.

[1] MERLN Institute for Technology-inspired Regenerative Medicine, Complex Tissue Regeneration Department, Maastricht University, Maastricht, The Netherlands. [2] POLYMAT, University of the Basque Country UPV/EHU, San Sebastián, Gipuzkoa, Spain. [3] IKERBASQUE, Basque Foundation for Science, Bilbao, Spain. ✉email: l.moroni@maastrichtuniversity.nl

The field of tissue regeneration emerged in the early 90's under the concept of designing active scaffold materials that can recapitulate several functions of the native targeted tissue[1], promoting the formation of neo-tissues, which have been so far limited to mechanical[2] and structural properties[3–5], and the implementation of bioactive factors[6–9]. However, cells within a tissue are subjected to dynamic (rather than static) stimuli that determine their function and fate[10]. Recent advances in scaffold design have thus been focused on tackling the temporal character of the native cell. Dynamic physical stimuli such as mechanical, sonic, magnetic or electrical stimuli on in vitro cell culture has proven effective in modulating multiple cell responses. However, extrapolation to in-vivo applications where the stimuli has to be externally applied has been limited[11–15]. Among these, ultrasound stimulation has found the most exploitation in clinical environments in detection[16] and therapeutic applications[17]. Lately, the use of ultrasounds for enhanced bone fracture healing has been studied both in vitro and in vivo with relative success, pointing at the need for more accurate systems that can efficiently transmit the applied waves. Moreover, the pulse of the ultrasound wave has also a great impact on the outcome of the stimulation[18]. On these studies, low intensity pulsed ultrasounds (LIPUS) with frequencies on the range of few MHz are directly applied to cells or to cells seeded on a construct that, at these frequencies, remain static[19–21]. Some examples of low frequency ultrasound stimulation of cell substrates have shown the relevance of these to induce osteogenic differentiation of cells, but this concept has not been exploited as functional 3D scaffolds, neither had the possibility of extrapolation to realistic in-vivo situations[22,23].

Additive manufacturing, commonly known as 3D printing, has shown great potential in tissue regeneration applications due to the ease of fabrication of patient-customized scaffolds.[24] Changes to the morphology of a 3D-printed object upon exposure to external triggers has been recently defined as 4D printing. Examples have so far been limited to one-way, non-reversible morphological transformations that are induced via permanent crosslinking of the materials, heat or pH changes non-suitable for physiological environments[25–27]. Few examples exist of morphological transformations that are two-way, reversible and repeatable, and these are based on the use of magnetic, inorganic and non-biocompatible materials that are not suitable for biological applications[28,29].

To bring further the field of tissue regeneration and render dynamic traditionally static 3D printed scaffolds, we demonstrate a concept based on the use of ultrasounds as stimulus. Under low frequency ultrasound stimulation the 3D printed scaffolds deform, becoming dynamic on-command when ultrasounds are activated, similarly to 4D printed objects. To control the extent and the pulse at which the scaffolds deform we took an engineering approach, designing scaffolds as ultrasound transducers with a deflecting and a damping element. We first developed an additive manufacturing strategy that induced phase segregation of biopolymer blends, which led to the formation of scaffolds with spatially controlled chemistries. At a 50:50 PLA:PCL biopolymer ratio, we induced the formation of a Janus-like phase that occurs homogeneously along the 3D structure. We modeled the mechanical deflection of the designed scaffold materials upon remote stimulation with different sound frequencies and investigated their effect on cell proliferation. Further, we investigated the deflection of Janus scaffolds that showed a shorter pulse width, behaving as transducers, and used them as biomaterial platforms to enhance bone formation. Our data shows that ultrasound stimulation of Janus scaffolds led to a more pronounced osteogenic differentiation, compared to non-stimulated (static) cultures on the same materials, with higher expression of osteogenic markers and higher deposition of the matrix proteins collagen I and fibronectin. Moreover, we found that this enhanced differentiation was connected to an increased formation and further activation of voltage-gated calcium ion channels, suggesting this biological pathway was activated.

## Results

**Fabrication of 3D structures with spatially controlled chemistries via phase segregation.** Additive manufacturing allows for the fabrication of 3D scaffolds with well-defined structures, but spatial control of the distribution of the deposited materials has so far been limited to the switch of materials between deposited layers (hundreds of microns) or to gradient structures at best[24]. To overcome this issue, we exploited basic concepts of physicochemistry, inducing phase-segregation of biopolymer blends (Fig. 1). We chose two widely used biodegradable and biocompatible polymers, polycaprolactone (PCL) and poly (D,L)-lactide (PLA), and blended them via extrusion to form filaments that were then used to feed a traditional fused deposition modeling (FDM) printer (Fig. 1a). Despite accounting for a similar Hildebrand solubility parameters (PCL 9.2 $(cal/cm^3)^{1/2}$ and PLA 10.1 $(cal/cm^3)^{1/2})$)[30] PCL and PLA have no favorable interactions and are therefore immiscible. Phase segregation of this polymer system has extensively been studied due to the favorable reduction of brittleness of PLA when blended with PCL[31–33]. Thus, initial blending to form filaments with PLA:PCL ratios of 20:80, 30:70, 40:60, 50:50, 60:40, 70:30, and 80:20 led to phase segregation (Supplementary Fig. 1). PCL was able to crystallize from the melt, while the PLA was amorphous (Supplementary Fig. 2). During the printing process, the polymer blends were molten and cooled down to room temperature (RT) while being deposited when PCL crystallized. Scanning electron microscopy images of the cross-section of the scaffolds showed that PCL and PLA phase-segregated during the printing process (Fig. 1b). A detailed observation of the formed phases via transmission electron microscopy (TEM, Fig. 1b) on the cross-sectional and longitudinal directions of the fibers of the scaffolds showed the clear formation of PLA phases (light gray) within a PCL-rich matrix (dark gray) at a PLA:PCL ratio of 20:80. At this polymer ratio, the phase segregation followed the principle of nucleation, with PLA median particle cross-sectional and longitudinal areas of 0.152 $\mu m^2$ (Confidence Interval (CI) 95.4%) and 0.216 $\mu m^2$ (CI 95.9%), respectively (Supplementary Figs. 3 and 4). Increasing the PLA:PCL ratio to 30:70 led to a spinodal decomposition and the formation of anisotropic phases that present maximum aspect ratios on the longitudinal (1.42; CI 95.6%) and transversal directions (2.77; CI 95.8%). At a PLA:PCL ratio of 50:50, the formation of well separated Janus structures was visualized by SEM, TEM, polarized light microscopy and light scanning microscopy (LSM), for which Rhodamine B and FITC were covalently attached to PCL and PLA, respectively, prior to scaffold fabrication (Fig. 1 and Supplementary Fig. 5). Moreover, the formation of Janus structures was observed to be homogeneous along the 3D structure, accounting for the same orientation of the phases along the depth of the structure (Fig. 1). The phase segregation on Janus fiber's cross-sections corresponded to the ratio of the two polymers on the blend, with an average occupied area of PLA phase of 48.2% ± 8.2 (±standard deviation) (Supplementary Fig. 6). At higher PLA:PCL ratios (60:40), the phases were inverted, forming PCL-elongated particles within a PLA matrix, with maximum particle sizes of 3.14 $\mu m^2$ (CI 95.3%) and 1.17 $\mu m^2$ (CI 96.01%) on the respective longitudinal and transversal directions. At the 80:20 PLA:PCL ratio, the particles became more spherical with aspect ratios of 1.3 (CI 95.1%) and 1.1 (CI 95.6%) on the longitudinal and transversal directions, respectively.

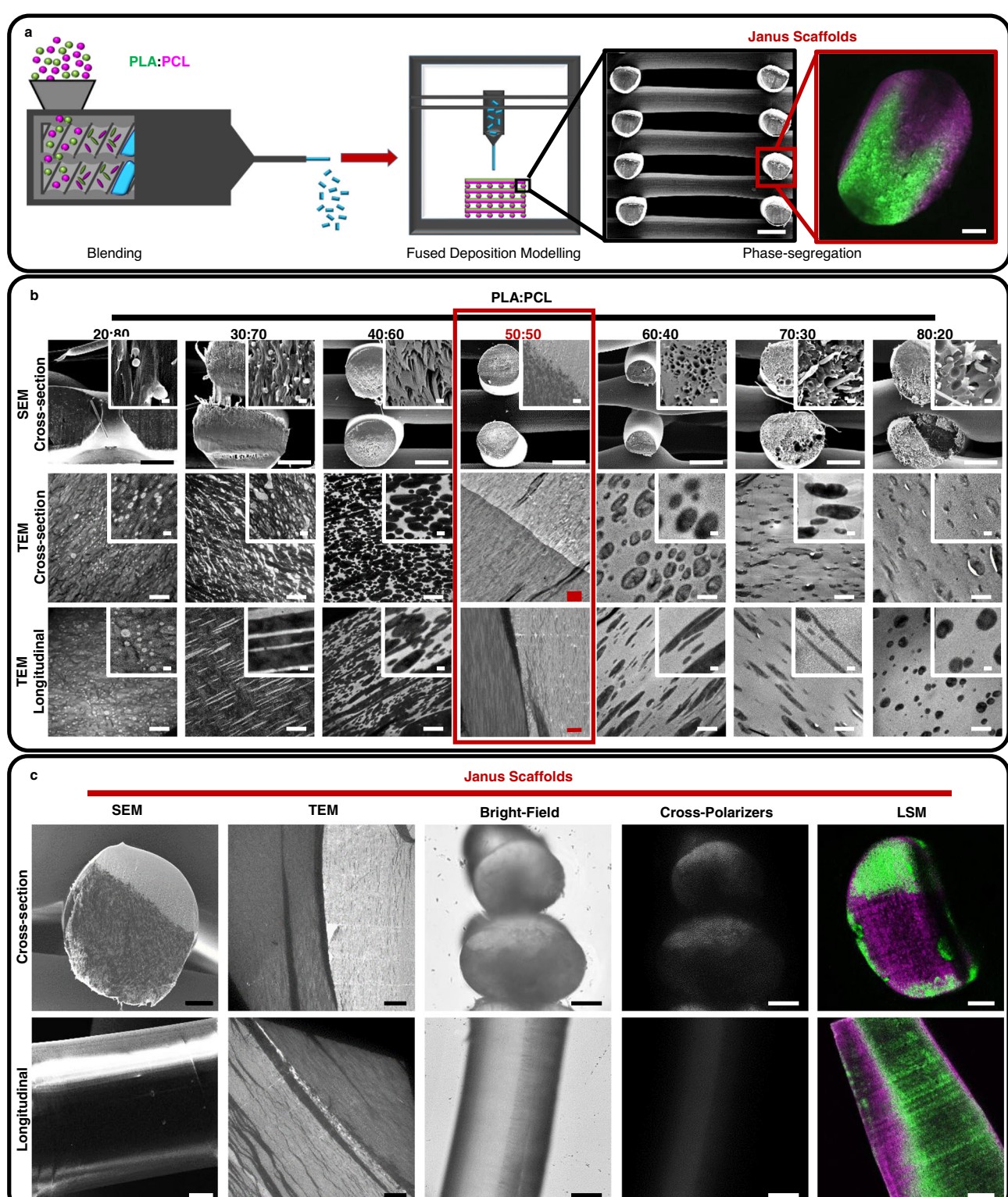

**Fig. 1 PLA:PCL blends phase-segregate during FDM following nucleation, growth and spinoidal decomposition and form Janus scaffolds at a 50:50 polymer ratio. a** Schematic representation of the process to create phase-segregated scaffolds. The materials were first blended to obtain filaments that were then cut in pellets to feed the 3D printer (left) where the phase segregation occurs as the material is deposited (middle) to give rise to Janus scaffolds (right). **b** Characterization of the phase-segregation in the cross-section of FDM scaffolds by scanning electron microscopy (SEM, scale bars are 500 and 5 μm for the insets), and transmission electron microcopy (TEM) on the cross-sectional and longitudinal directions. PCL appears as high-contrast phases (crystalline, dark) and PLA as low contrast phases (amorphous, bright). For the TEM images, white scale bars are 5 μm and 1 μm on the insets; red scale bars are 10 μm. Red box indicates Janus scaffolds. **c** Characterization of Janus structures in the scaffold's fiber cross-sectional and longitudinal directions by SEM, TEM, optical microscopy on bright-field and under cross-polarizers, and light scanning microscopy (LSM). Scale bars are 100 μm except on TEM images were scale bars are 10 μm.

From these data, it appeared evident that phase segregation occurs during printing and that different polymer ratios led to definite structures, including a Janus structure at 50:50 PLA: PCL ratio.

**Exploitation of Janus scaffolds as ultrasound-activated, dynamic mechanical scaffolds**. Having formed Janus scaffolds with distinct phases, we wished to investigate whether they could be exploited to induce mechanical stimulation to cells using biologically relevant stimuli. We selected ultrasounds at different frequencies to evaluate the optimal deflection of the scaffolds that could trigger a cell response. We modeled the deflection of single-layered Janus, PCL and PLA scaffolds fixed on their extremes (mimicking implantation) and in liquid media under the application of continuous sound waves at different frequencies (Fig. 2b). The frequencies and wave intensities were defined from experimental measurements of our homemade ultrasound set-up (Supplementary Figs. 7, 8, 9 and Supplementary Table 1). The speed of sound on materials with similar densities, depends on the bulk stiffness of the media as per Eq. (1):

$$\nu = \sqrt{\frac{E}{\rho}} \qquad (1)$$

where $\nu$ is the speed of sound, $E$ is the bulk Young's modulus of the media and $\rho$ is the density. At the same time, the speed of the wave is defined by Eq. (2)

$$\nu = \lambda \cdot f \qquad (2)$$

where $\lambda$ is the wavelength and $f$, the frequency of the wave. Thus, for a given stimulation frequency, the wavelength of the transmitted wave (after crossing a material) will increase with increased material stiffness. Therefore, the flexural modulus of the materials was measured under 3-point bending and used for the computational modeling of the scaffold deflection under ultrasound stimulus. The flexural modulus was $759 \pm 62$ MPa, $303 \pm 70$ MPa and $2326 \pm 36$ MPa for Janus, PCL and PLA, respectively (Supplementary Fig. 10 and Supplementary Table 2). For the selected frequencies of 10, 20, and 40 kHz, deflection amplitudes were determined: PLA displayed maximum deflections of 711, 112, and 20.7 nm, followed by Janus with 207, 79.3, and 19.8 nm and PCL scaffolds with 142, 47.1, and 14.2 nm for frequencies of 10, 20, and 40 kHz, respectively. These data indicate that deflection of the sound waves was attenuated at higher sound frequencies and by softer materials such as PCL. Knowing the predicted deflection amplitude for a given material and at the different sound frequencies, we then investigated its effect on hBMSCs proliferation.

We seeded hBMSCs on the different scaffolds and cultured them for 7 days applying 30 min of stimulation daily at either 0, 10, 20, or 40 kHz using the same structures as modeled in Fig. 2b (Fig. 2c, d, e and Supplementary Fig. 11), following the duration of previously reported ultrasound stimulation protocols[19,21]. At sound frequencies of 10 kHz when all PCL, PLA, and Janus scaffolds displayed the greatest deflection, the cell number decreased, as compared to their respective counterparts in static culture (0 kHz). At ultrasound stimulation frequency of 20 kHz, a slight increase in the proliferation relative to non-stimulated cultures was measured for cells cultured on PCL and Janus scaffolds, which was more pronounced in PLA materials. Thus, lower deflection amplitudes (higher frequencies) resulted on higher proliferation relative to non-stimulated cultures when the materials were analyzed individually. The normalized proliferation was not only dependent on the amplitude of the deflection, but also on the material used. The deflection amplitude of PCL at 10 kHz was of 142 nm, comparable to the deflection of PLA when stimulated at 20 kHz, 112 nm.

However, the normalized proliferation on PCL scaffolds at 10 kHz was $26 \pm 8\%$ while a proliferation of $408 \pm 185\%$ was measured for hBMSCs cultured on PLA scaffolds at 20 kHz (Fig. 2d and Supplementary Fig. 11). Similarly, ultrasound stimulation at 40 kHz, which resulted on the smallest deflection in all the materials, led to an the highest increased cell proliferation in PLA and Janus scaffolds when normalized to non-stimulated cultures, with cell densities of $4492 \pm 341$ and $5219 \pm 555$ cell.cm$^{-2}$ and normalized proliferation of $519 \pm 111$, $234 \pm 70\%$, respectively. However, hBMSCs normalized proliferation in stimulated PCL scaffolds was not significantly different to its counterpart in static conditions (0 kHz), with a cell density of $1044 \pm 97$ cell.cm$^{-2}$ and a normalized proliferation $89 \pm 18\%$, indicating that ultrasound stimulation was not effective in PCL materials. To decipher what was the reason for this differential behavior we further investigated the effect of ultrasound stimulation at 40 kHz on cell matrix deposition and differentiation.

Culture of hBMSCs on the scaffolds for 7 and 14 days under ultrasound stimulation (30 min/day at 40 kHz) resulted in the formation of a fibronectin-rich ECM on Janus scaffolds, while cells on PLA scaffolds and on static cultures presented only intracellular expression of the protein (Fig. 2f and Supplementary Figs. 12 and 13). hBMSCs on PCL deposited fibronectin with or without stimulation and populated the scaffold with a reduced cell number, indicating again that ultrasound stimulation was not effective in this particular material. Cells cultured in PLA and stimulated at 40 kHz had a higher proliferation rate, but their capability to deposit fibronectin was unaltered (Figs. 2f and S12). Cells cultured in Janus scaffolds, on the other hand, responded with an increased proliferation rate and matrix deposition when stimulated at 40 kHz compared to static cultures. We therefore hypothesized that despite the similar deflection amplitude modeled for the different materials, these must had a different response in the pulse of the deflection that affected the cell response, as previously shown[18].

**Janus scaffolds are a combination of a deflecting and a damping material**. Ultrasound transducers are fabricated as a combination of a damping material (dissipating the energy and remaining static or with very little vibration) that is placed underneath the active or vibrating material (storing energy). The combinatorial response of the sandwich composite resulted in a reduced pulse length or width (and amplitude) of the transmitted wave (Fig. 3, schematic). Similarly, Janus scaffolds accounted for a PCL phase with a higher energy dissipation or damping potential (tangent δ at 25 °C and 0.1 Hz of $11.3 \times 10^{-2} \pm 0.4 \times 10^{-2}$) than the PLA phase (tangent δ at 25 °C and 0.1 Hz of $6.3 \times 10^{-2} \pm 0.9 \times 10^{-2}$), which presented a higher energy storage potential, with a storage and loss moduli of $42.8 \pm 5.9$ MPa and $4.9 \pm 0.5$ MPa for PCL, and $81.6 \pm 13.7$ MPa and $5.4 \pm 1.1$ MPa for PLA (Supplementary Figs. 14 and 15). The resulting Janus accounted for a storage modulus of $89.5 \pm 12.2$ MPa, a loss modulus of $6.8 \pm 0.3$ MPa and a tangent δ of $7.6 \times 10^{-2}$ $0.4 \times 10^{-2}$.

To study the potential differences on pulse width and pulse repetition frequency (PRF), we first measured the wave transmitted by PLA, PCL and Janus scaffolds when stimulated at 40 kHz in liquid media with the use of a hydrophone detector (Supplementary Figs. 7a, 16, 17 and Supplementary Table 3). Applying Fast Fourier Transform (FFT) to the recorded data, the response was decoupled onto a wave with the characteristic frequency of the input signal (approximately 40 kHz) and a secondary wave (the pulse) that was dependent on the material used. Next, we measured the deflection of the scaffolds upon ultrasound activation at 40 kHz with a nanoindenter (Figs. 3 and S7, b). The tip of the cantilever was approached to the surface and

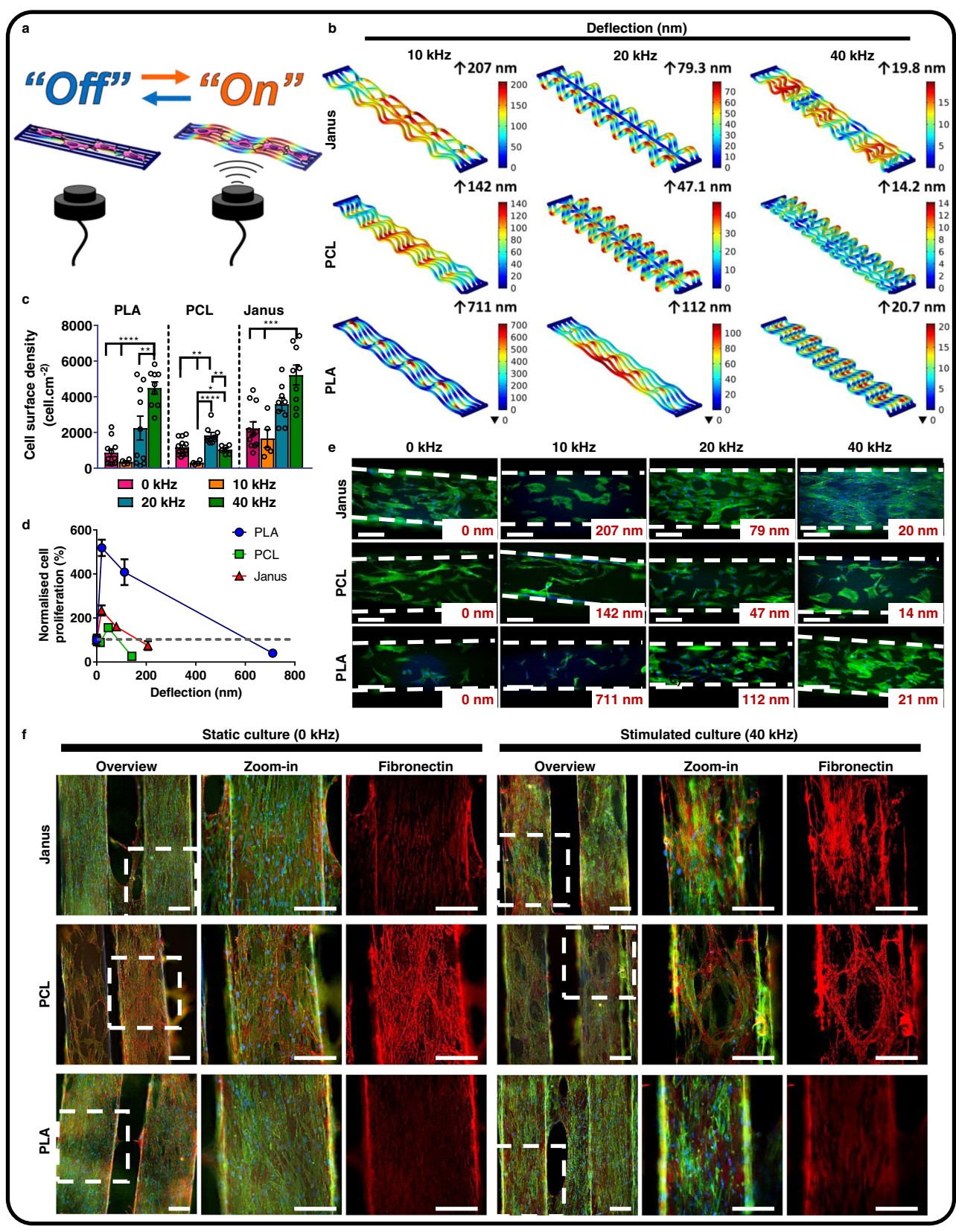

engaged (500 nm indentation), after which the ultrasound was switched on and the response on the deflection of the scaffolds was measured (Fig. 3, indentation). PLA scaffolds deflected with a pulse amplitude of 53 ± 2 nm, a pulse width of 2 cycles (0.49 s) and a PRF of 1.1 Hz. PCL scaffolds, however, barely reacted and behaved as damping materials, with a response that was not pulsed (rather sinusoidal) and with an amplitude of deflection of 4 ± 1 nm. The combination of these two materials, PLA as active and PCL as damping material, in Janus scaffolds resulted in a reduced amplitude of 36 ± 4 nm and pulse width (1 cycle, 0.1 s), and an increased PRF of 2.17 Hz compared to PLA scaffolds, as it occurs on ultrasound transducers. The amplitude of the

**Fig. 2 FDM scaffolds respond to ultrasound stimulation affecting cell proliferation. a** Schematic representation of the set-up used to stimulate hBMSC cultures showing fixed FDM scaffolds and remote ultrasound stimulation. **b** Computational simulation of the deflection of the different scaffolds (Janus, PCL and PLA) under sound stimulation at 10, 20, or 40 kHz. Color scale represents deflection in nm. **c** Quantification of cell surface density in **e** (For stimulation at 0, 10, 20, and 40 kHz, $n = 11$, 4, 10, and 10 for PLA; $n = 13$, 4, 10, and 6 for PCL and $n = 11$, 5, 9, and 9 for Janus, respectively) Data is shown as means ± SEM and represents images from 3 biological triplicates. Statistical significance was calculated by two-way ANOVA with Tukey's multiple comparison test; ****$p < 0.0001$, **$p < 0.01$, and *$p < 0.1$. Source data and exact $p$ values are provided as a Source Data file. **d** Cell proliferation as a function of the simulated scaffold deflection (nm) from measurements in **c**. Data is shown as means ± SEM. Black circles are individual data points. Dashed line represents cell number in hBMSC culture with 0 kHz stimulation. **e** Fluorescence microscopy images of hBMSCs cultured in Janus, PCL and PLA scaffolds and stimulated for 30 min at different frequencies for 7 days showing an ultrasound-dependent and material-dependent cell proliferation/density, $n = 3$. Labels indicate beam deflection as calculated in **b** and white dashed line define the limits of the fiber. Cells were stained for F-actin (green) and DNA (blue). Scale bars represent 200 μm, $n = 3$. **f** Fluorescent light microscopy images of hBMSCs cultured for 14 days on Janus, PCL and PLA scaffolds and stimulated for 30 min daily at 40 kHz. Cells were stained for F-actin (green), DNA (blue), and fibronectin (red). Scale bars represent 200 μm. White dash-line boxes represents the area where the zoom-in was taken.

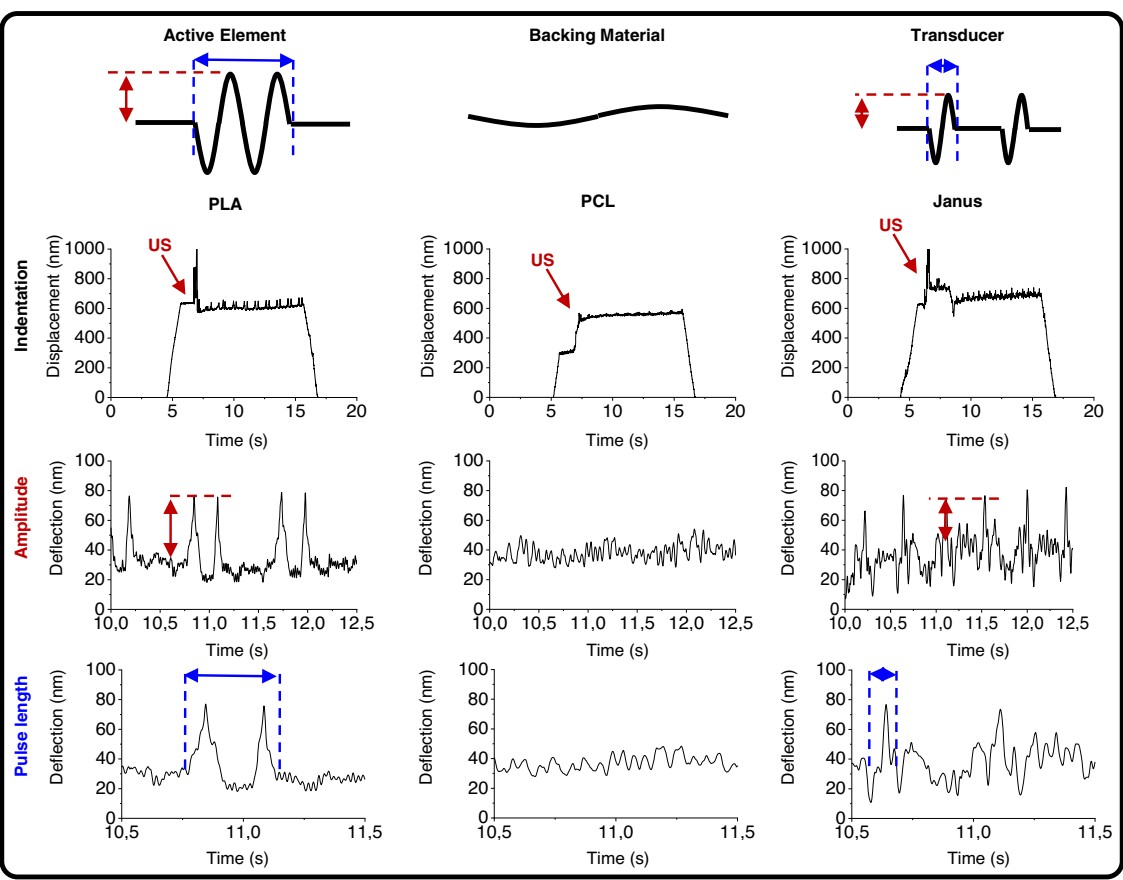

**Fig. 3 Janus scaffolds respond to ultrasound stimulation as transducers composed of a reactive material (PLA) and a backing material (PCL), and affect matrix deposition by cells.** Schematic representation of the response of the different components of a transducer element (top). Graphs of indentation experiments (bottom) on PLA, PCL and Janus scaffolds showing a decrease of the amplitude (red arrows) and pulse length (blue arrows) on Janus scaffolds as compared to PLA scaffolds upon ultrasound stimulation at 40 kHz (red arrows labeled US).

deflection, measured by nanoindentation correlated well with the trend observed on the models showing higher deflection for PLA, followed by Janus and PCL scaffolds. The difference on the absolute values of the deflection probably arise from the result (sum) of the incident and echoing waves not contemplated on the model, resulting in greater deflection amplitudes. The differences we observed in cell proliferation might be related to the pulse of the scaffold deflection, being the pulse width highest in PLA scaffolds, followed by Janus and PCL scaffolds. It is noteworthy that the PCL response to the applied 40 kHz frequency resulted on a continued rather than pulsed wave. We hypothesized that the continuous mechanical stimulation, although smaller in amplitude might result on a stress to cells rather than a sudden

mechanical stimulation, thus reducing (or at least maintaining) the proliferation rate and not affecting the cell matrix deposition.

**Janus scaffolds promote osteogenic differentiation of hBMSCs under remote ultrasound stimulation.** Ultrasound-mediated external activation of scaffolds has mostly been reduced to the use of piezoelectric composite materials with susceptible particles, or of crystalline piezopolymers that result in an electrical signal[34]. However, the mechanical activation of scaffolds for tissue regeneration purposes via ultrasounds has yet not been investigated. To evaluate the potential of external ultrasound stimulation of 3D printed scaffolds for bone regeneration, we cultured

hBMSCs on Janus, PCL and PLA scaffolds for 21 days in osteogenic media and under static (0 kHz) and stimulated (40 kHz, 30 min/day) conditions. After 21 days of culture, a dense collagen I network was formed on stimulated Janus scaffolds (Fig. 4a) while stimulated PCL, PLA or any of the static culture conditions showed only intracellular expression. An upregulation of collagen I, RunX2 and osteocalcin was also detected at a gene level on cells cultured on stimulated Janus and PCL materials (Fig. 4b) compared to stimulated PLA or any of the static conditions, with the increase being more pronounced on Janus scaffolds. The cell number was also significantly increased on Janus and PLA materials with respect to their counterparts in static culture, in agreement with earlier observations (Fig. 4c and Fig. 2c). Alizarin red staining for $Ca^{2+}$ deposits was maximum on PCL and Janus scaffolds in stimulated culture (Fig. 4d). However, a detailed observation of the deposited minerals by SEM revealed that hBMSCs cultured on PCL underwent what has been proposed as chemical differentiation, depositing minerals with a smooth and big crystal-like morphology characteristic of $Ca^{2+}$ salts[35]. Minerals deposited on Janus scaffolds, however, showed a rounded and porous morphology typically ascribed to amorphous hydroxyapatite or calcium phosphate[35,36]. During bone mineralization, amorphous and globular shaped calcium phosphates undergo mineralization to form carbonated hydroxyapatite, thus suggesting the bone formation potential on Janus scaffolds[37–40]. Osteocalcin release increased with ultrasound stimulation for cells cultured on Janus and PLA scaffolds but not for PCL, with cells cultured on the last having the lowest release (Supplementary Fig. 18). Contrary to this, and as shown previously[41], alkaline phosphatase activity was similar for cells with and without ultrasound stimulation but highest for cells cultured on Janus scaffolds (Supplementary Fig. 19). ATP release, a key regulator of osteoblast response upon mechanical stimulation, was 4-fold higher in hBMSCs cultured on stimulated Janus scaffolds compared to any of the other stimulated or static conditions. ATP release has also been shown to increase upon direct low-intensity ultrasound stimulation, and consequent depolarization of the cell membrane, during osteogenic differentiation of MC3T3-E1 cells and in 2D cultures of mesenchymal stem cells, which is in agreement with what we observed here (Fig. 4e)[41,42].

**Osteogenic differentiation on stimulated Janus scaffolds occurs via activation of voltage-gated $Ca^{2+}$ ion channels (VGCC).** Cell membrane depolarization, as a consequence of mechanical stimuli, results in the activation of voltage-gated $Ca^{2+}$ ion channels (VGCC) to regulate calcium influx into the cell. Some reports have shown the presence of L-type VGCC in hBMSCs and suggested that they play a pivotal role in cell attachment, proliferation and osteogenic differentiation[43,44]. To determine whether L-VGCCs were involved in the osteogenic differentiation, we evaluated the expression of *CACNA1c* gene that encodes $Ca_v1.2$, a subunit of L-VGCC; we found that *CACNA1c* was upregulated 3-fold on stimulated Janus scaffolds but not on stimulated PCL or PLA scaffolds (Fig. 5a). L-type VGCC are activated via physical coupling of the $Ca_v1.1$ subunit of the dehydropyridine receptor (DHPR) to the Ryonodine receptor (RyR) on the endoplasmic reticulum of cells[45]. Staining of DHPR revealed the formation of L-VGCC on cells cultured in PCL and Janus scaffolds, and only the coupling of these to RyR on cells cultured on Janus scaffolds (Fig. 5b and Supplementary Fig. 20), proving the direct effect of mechanical deflection on Janus scaffolds via ultrasound stimulation. Indeed, blocking of L-VGCC with 1 μM nifedipine during the differentiation process resulted in a decreased cell number in all materials and culture conditions, but was more pronounced in stimulated conditions. The difference in cell number between

stimulated and static conditions was not significant in PCL and PLA scaffolds. However, the cell number measured in stimulated Janus scaffolds was significantly lower to their counterparts in static culture (Fig. 5c). Blocking of L-VGCC also resulted in the downregulation of collagen I, RunX2 and osteocalcin gene expression, resulting in no significant differences between stimulated and static culture conditions (Fig. 5d). Thus, when L-VGCC were blocked, ultrasound stimulation no longer influenced cell differentiation or proliferation, proving their direct correlation.

Here, we show two alternative routes to 4D printing: in-situ phase segregation to control spatially the composition of the printed structure, and ultrasound stimulation to remotely activate the deflection of the scaffolds. Varying the ratio of the polymer blend allows controlling the phase formation to give rise to particles, ellipsoidal phases or Janus structures that are spontaneously formed during the printing process. The formation of such phase-segregated structures provides additional control for 3D printing strategies, developing the technology towards 4D printing. Particularly, the control over the chemistry could be exploited for the selective functionalization of the different formed phases and thus further control the cell phenotype. Moreover, Janus scaffolds present reversible morphological changes upon ultrasound stimulation, that can be activated and de-activated on-command, which further defines them as 4D printed scaffolds. PCL and PLA scaffolds respond to ultrasound as damping and deflecting materials, respectively, and combinations of these as Janus structures results in shorter pulse widths and smaller deflection. These properties of the scaffolds directly affect cell proliferation, matrix deposition and osteogenic differentiation of cultured hBMSCs. Ultrasound stimulation of PCL scaffolds shows no significant effect in cell proliferation or matrix deposition, but a small increase in gene expression of some osteogenic markers and the deposition of $Ca^{2+}$ salts, ascribed to a chemical differentiation. Stimulation of PLA scaffolds results in an increased cell proliferation, but no effect on cell differentiation. Stimulation of Janus scaffolds affects hBMSCs with increased cell proliferation, and expression and deposition of osteogenic markers. We further show that L-VGCC are activated on cells cultured in stimulated Janus scaffolds, and that blocking these cancels the cellular effect of ultrasound stimulation. Taken altogether, we propose that remote activation of Janus scaffolds presents an ideal alternative to traditional static implants, providing on-command stimulation of cells. However, their applicability on in-vivo situations is yet to be investigated. Moreover, common sterilization processes such as ethylene oxide or gamma irradiation would need to be tested for their potential structural damage to the polyesters used herein.

## Methods

**Fabrication of scaffolds.** PCL and PLA were purchased from Sigma-Aldrich and NatureWorks, both with a molecular weight of 80 kD. The two polymers were blended at the defined ratios using a twin-extruder working at 150 °C and 100 rpm. The filaments were collected, chopped down to approximately 5 mm height and used to feed a fused-deposition modeling Bioscaffolder SYSENG at a feed rate of 500 mm/min and a dispensing volume of 30 rpm. The structure had a length of 30 mm, a strand distance of 0.6 mm and a layer thickness of 0.25 mm using a needle of 0.4 mm inner diameter. For cell culture experiments, single-layered scaffolds were glued from the extremes to petri dishes with Norland optical adhesive 68 and cured by 30 s of 395 nm UV exposure.

**Imaging of phase segregation by SEM, TEM, and POM.** For SEM and polarized optical microscopy (POM), scaffolds were cut with a razor blade perpendicularly to the sample height and directly imaged with a Nikon Ti-S with crossed polarizers. Samples for SEM were sputter-coated with gold (Cresignton 108 Auto) and imaged on a FEI/Philips XL-30 microscope under both secondary and backscattered electrons with an acceleration voltage of 10 keV. For TEM, samples were embedded on Epon LX112 resin (Hexion) and slowly cured at RT for 21 days. Blocks were then sliced on an ultra-cryomicrotome Leica EM FC6 at temperatures of −100 °C

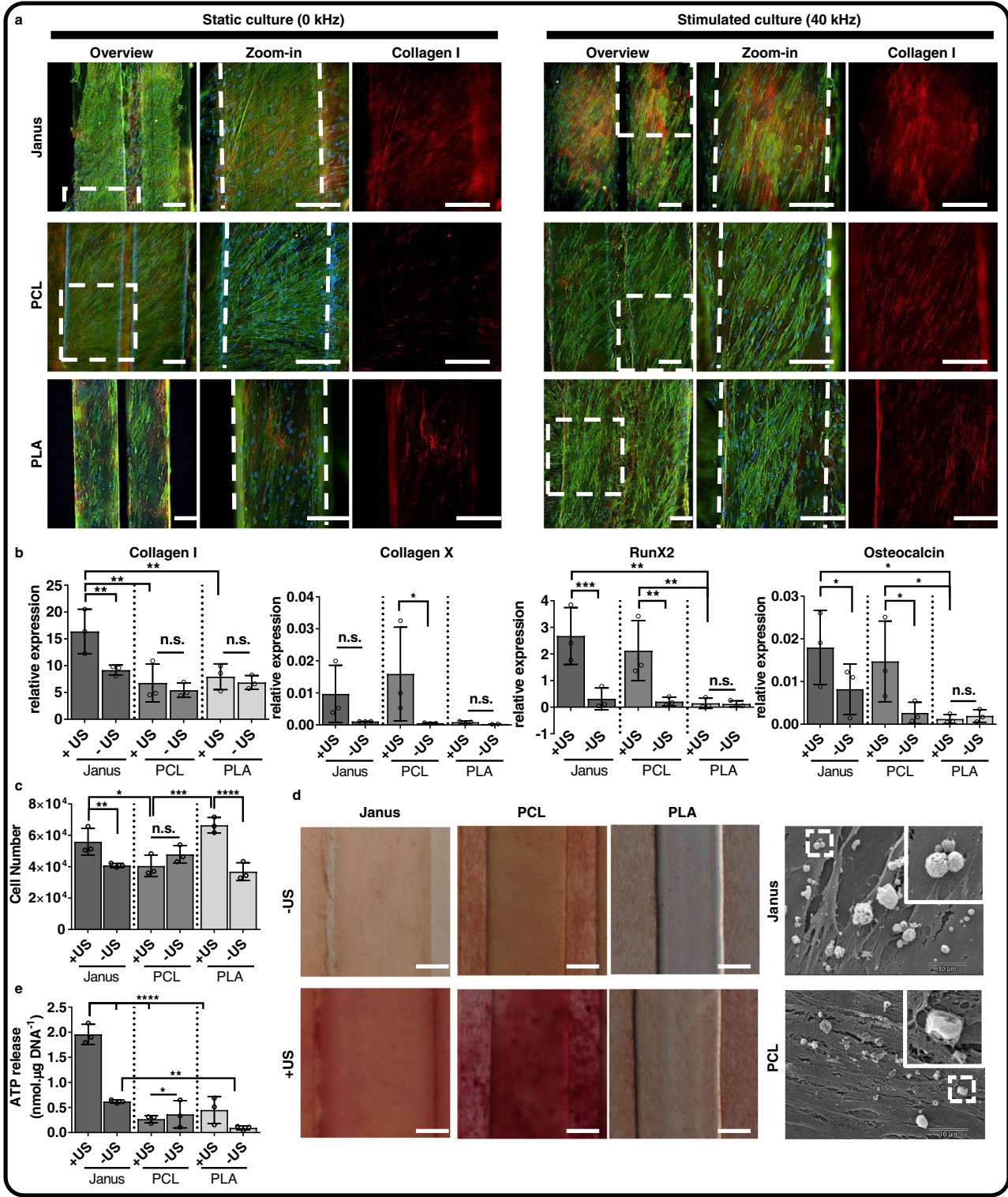

**Fig. 4 Osteogenic differentiation is enhanced on cells cultured for 21 days on ultrasound-stimulated Janus scaffolds.** hBMSCs were cultured for 21 days on Janus, PCL and PLA scaffolds in osteogenic media and under static (0 kHz, −US) or dynamic conditions (40 kHz, 30 min/day, +US) after which (**a**) collagen I deposition was evaluated from fluorescence light microscopy images. Cells were stained for F-actin (green), DNA (blue) and Collagen I (red). White dash-line boxes represents the area where the zoom-in was taken and dash lines define the limits of the fibers. Scale bars represent 200 μm. **b** Collagen I, collagen X, RunX2, and osteocalcin gene expression and cell number (**c**) were also analyzed. **d** Alizarin red staining (left) showing increased calcium deposition on Janus and PCL scaffolds stimulated with ultrasound compared to static cultures. SEM images (right) showing the characteristic morphology of hydroxyapatite on Janus scaffolds, while deposits on PCL substrates represent $Ca^{+2}$ salts. White dash-line boxes represents the area where the zoom-in was taken. Scale bars represent 100 μm. **e** Measurement of the ATP on hBMSCs cultured on Janus, PCL and PLA scaffolds. In all the graphs, data is shown as mean ± SD. Black circles are individual data points. $n = 3$ biological triplicates. Statistical significance was calculated by ordinary two-way ANOVA with corrected Tukey's multiple comparison test between groups and uncorrected Fisher's LSD between conditions of a same group (+/− US); ****$p < 0.0001$, ***$p < 0.001$, **$p < 0.01$, and *$p < 0.1$. Source data and exact $p$ values are provided as a Source Data file.

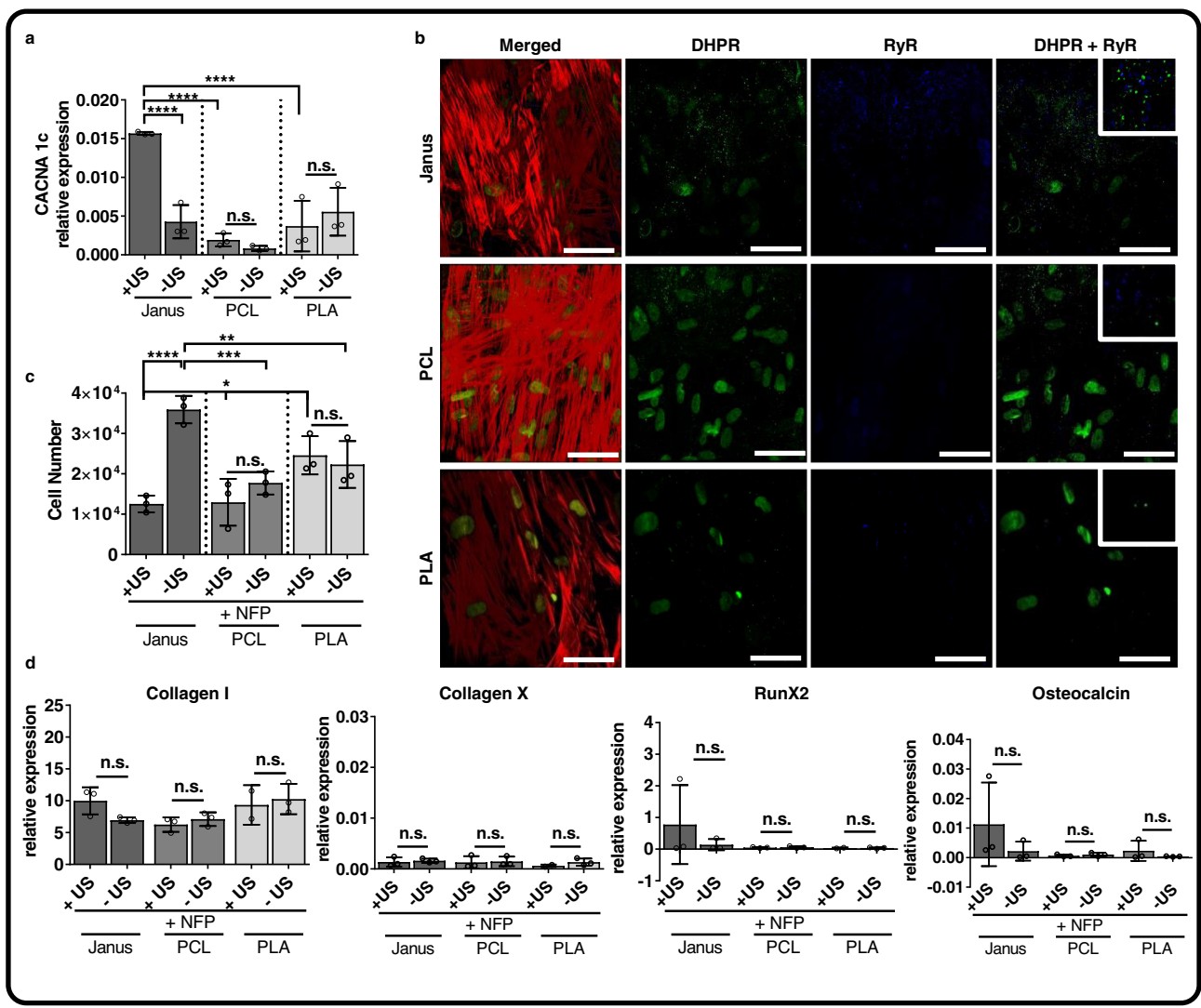

**Fig. 5 Enhanced osteogenic differentiation occurs via formation and activation of voltage-gated Ca$^{2+}$ ion channels.** hBMSCs cultured for 21 days in osteogenic media on Janus, PCL and PLA scaffolds under stimulated (+US, 40 kHz) and static (−US, 0 kHz) conditions showed (**a**) highest CACNA1c (L-type voltage-gated Ca$^{2+}$ ion channel) gene expression on stimulated Janus scaffolds. **b** Light scanning microscopy images revealed the presence of dihydropyridine receptor (DHPR, a voltage-gated Ca$^{2+}$ ion channel) and coupling with Ryonodine receptor (RyR) only on cells cultured on Janus scaffolds. Cells were stained for F-actin (red), DHPR (green) and RyR (blue). Scale bars represent 50 µm and insets are 18 µm. Culture of hBMSCs for 21 days in osteogenic media were L-type voltage-gated Ca$^{2+}$ ion channels were blocked with 1 µM nifepidine (+NFP) showed an overall decrease in cell number (**c**). **d** Gene expression of osteogenic markers collagen I, collagen X, RunX2, and osteocalcin, decreased in all materials and culture conditions when nifepidine was used and showed no significant difference between dynamic and static cultures. In all graphs, data is shown as mean ± SD and $n = 3$ biological triplicates. Black circles are individual data points. Statistical significance was calculated by two-way ANOVA with corrected Tukey's multiple comparison test between groups and uncorrected Fisher's LSD between conditions of a same group (+/− US); ****$p < 0.0001$, **$p < 0.01$, and *$p < 0.1$. Source data for the graphs and exact $p$ values are provided as a Source Data file.

and −45 °C for the chamber and for the knife, respectively. The samples were collected on TEM carbon-supported Cu grids from the 60% DMSO bath. The grids were imaged on a TEM FEI/Tecnai G2 Spirit BioTWIN at a typical acceleration of 80 keV.

**Computational models of scaffold deflection.** Simulations of the scaffolds' deflection under different sonication frequencies was performed using COMSOL Multiphysics modeling software to compute the pressure variation for propagation of acoustic waves in the scaffolds when submerged on a cylinder of aqueous media at quiescent background conditions. As input parameters, the density of the materials and the calculated flexural modulus were used. The applied wave sounds were circular, with an intensity of 60 Pa and at defined frequencies of 10, 20, or 40 kHz.

**Ultrasound stimulation and detection set-up.** A Kemo M048N ultrasound generator (Kemo Electronic) with a frequency range of 2–40 kHz was connected to four Kemo P5123 mini piezoelectric tweeters (Kemo Electronic) with frequency

range of 2.5–45 kHz and a pressure of 118 dB. The petri dishes containing the scaffolds were place on top of the loud speakers that were mounted on flat surfaces (Supplementary Fig. 7). To characterize the waves after transmission through the petri dishes (empty controls and containing scaffolds) and in aqueous media, a 0.2-mm needle hydrophone (NH0200, Precision Acoustics) was connected to a hydrophone preamplifier and a DC coupler that were then connected to a Rigol DS1102 oscilloscope. Collected data was plotted and analyzed using Origin 2018 software.

**Mechanical properties of the materials.** The flexural modulus of PCL, PLA and 50:50 blends was measured under 3-point bending from extruded filaments of 2 mm diameter mounted on a 1 cm support spam. A TA ElectroForce (TA Instruments) mechanical tester equipped with a 45 N load cell and controlled with WinTest DMA 7.1 software was used to deform he samples at a strain rate of 0.01 mm/s. The experiments were run until creep (or until the maximum applicable load was reached). The flexural modulus was calculated from the slope of the linear region of recorded force-displacement curves. The storage and loss moduli

**Table 1 List of primers used for RT-PCR.**

| Gene | Forwards primer 5′ to 3′ | Reverse primer 3′ to 5′ |
|---|---|---|
| COL1a1 | AGGGCCAAGACGAAGACATC | AGATCACGTCATCGCACAACA |
| COLXa1 | GAC TCC CTC CTC ACT GTC GC | AGG GAA GTC TCC CTC ACT TGT |
| RUNX2 | AGT GAT TTA GGG CGC ATT CCT | GGA GGG CCG TGG GTT CT |
| OCN | TGAGAGCCCTCACACTCCTC | CGCCTGGGTCTCTTCACTAC |
| CACNA1c | GAAGCGGCAGCAATATGGGA | TTGGTGGCGTTGGAATCATCT |

and the tangent δ of PCL, PLA and Janus scaffolds were measured from 3D printed single fibers of 400 μm diameter using a TA Q800 dynamic mechanical thermal analyzer. Experiments were run with a thermal sweep from 20–65 °C, a 3 °C temperature ramp, a dynamic strain of 0.5% and fixed frequency of 0.1 Hz from which the storage modulus, loss modulus and tangent δ are reported as mean ± SD, $n = 3$. Experiments were also run under a frequency sweep of 0.1–100 Hz at a fixed temperature of 37 °C using a strain of 0.5%. All PLA samples broke above 1.6 Hz. $n = 3$.

**Measurement of scaffold deflection**. The deflection of the scaffolds was measured in a Piuma nanoindenter (Optics11) with a MN2-TN1-ST probe of 0.5 N/m spring constant (Optics11) and in aqueous media using the same sonication set-up described above. The samples were visualized with the built-in camera, and the cantilever probe was brought to the center of one of the scaffold fibers. After engaging, an indentation maximum 500 nm was applied and held for 10 s. Immediately after indentation, an ultrasound frequency of 40 kHz was applied (Supplementary Fig. 7). Collected data was plotted and analyzed using Origin 2018 software.

**Ethics declarations**. hBMSCs were obtained from the Center for the Preparation and Distribution of Adult Stem Cells in accordance with procedures approved by the Scott and White and Texas A&M Institutional Review Boards. Bone marrow aspirates were retrieved from healthy donors after informed consent. After collection, bone marrow was centrifuged to isolate mononuclear cells, which were then plated, expanded and analyzed for their differentiation potential into the three lineages. The Center for the Preparation and Distribution of Adult Stem Cells has distributed cells to over 250 laboratories worldwide through the grant #P40RR017447 from NCRR of the NIH.

**Cell culture**. hBMSC (22-year old, male) were kindly provided by Texas A&M Health Science Center of Medicine Institute for Regenerative Medicine at Scott and White[46]. Cryopreserved vials at passage 2 were plated at a density of 1000 cells/cm² and cultured in basal media consisting of alpha-MEM media supplemented with Glutamax (Gibco) and 10% fetal bovine serum (FBS) (Sigma-Aldrich). Cells were subcultured at 80% confluence. All experiments were performed at cell passage 5.

**Cell culture in 3D printed scaffolds under sonic stimulation**. 3D printed scaffolds were sterilized in 70% ethanol, rinsed with PBS and coated for 1 h with human recombinant vitronectin (Thermo Fisher) at a surface density of 1 μg/cm⁻² and assuming 3 cm² per sample.

hBMSCs were seeded at a density of 15,000 cell/cm² from cell-concentrated dispersions of $1.5 \times 10^6$ cell/mL (30 mL) on top of the scaffolds and incubated for 2 h, after which 3 mL of media were added. Media was refreshed every day right after the 30-min ultrasound stimulation. Cell proliferation experiments were performed in basal media supplemented with 100 U/mL penicillin-streptomycin, and cells were stimulated 30 min/day for 7 days, after which samples were taken for analysis. Matrix deposition experiments were performed in basal media supplemented with 100 U/mL penicillin-streptomycin and 0.2 mM L-ascorbic acid. Samples were stimulated daily for 30 min at 40 kHz for 14 days. After 7 and 14 days, samples were harvested for analysis. For osteogenic differentiation experiments, hBMSCs were cultured in osteogenic media composed of alpha-MEM media supplemented with Glutamax (Gibco), 10% FBS, 100 U/mL penicillin-streptomycin, 0.2 mM L-ascorbic acid, 100 nM dexamethasone and 10 mM β-glycerophosphate. To block L-type VGCC, 1 μM nifedipine was added to the osteogenic media. Cells were stimulated for 30 min/day at 40 kHz and cultured for 21 days, after which samples were taken for analysis. All experiments were performed with non-stimulated (or static culture, 0 kHz) controls. All the experiments and analysis were performed in triplicate, $n = 3$.

**Immunofluorescence**. Samples were fixed in 4% paraformaldehyde for 25 min and permeabilized 15 min in a 0.1% Triton X-100 solution in PBS. Samples were then blocked for 1 h at RT in a solution of 3% bovine serum albumin (BSA) and 0.01% Triton X-100 in PBS. After rinsing with PBS, samples were incubated 1 h at RT with primary antibody: rabbit anti-fibronectin (1:400, Abcam ab2413), mouse anti-collagen I (1:400, Abcam ab90395), mouse anti-DHPR alpha 2 subunit/

CACNA2D1(1:200, Abcam ab2864) or rabbit anti-Ryanodine (1:100, Abcam ab219798). Samples were then rinsed with a solution of 0.3% BSA and 0.001% Triton X-100 in PBS and incubated for 1 h at RT with Alexa Fluor–conjugated secondary antibodies (1:200). F-actin and DNA were stained incubating the samples 1 h with Alexa Fluor 488-phalloidin or Alexa Fluor 568-phalloidin (1:100) followed by thorough rinsing and 15 min with 1:1000 (wt:vol) Hoechst 34580 trihydrochloride salt (Sigma-Aldrich). hBMSCs were imaged on a Nikon TE2000 PFS fluorescence microscope. Visualization of DHPR and RyR stains was also performed on a Leica TCS SP8 CARS. Quantification of cell number was done using CellProfiler free software. Quantification of the area covered by fibronectin was done using FiJi free software.

**Gene expression analysis**. After 21 days of culture with static and stimulated conditions, samples were harvested and subjected to 3 freeze-thaw cycles. Total RNA isolation was carried out on a RNeasy Minikit with on column DNase treatment (Quiagen), according to manufacturer's protocol. cDNA was synthesized using iScript cDNA synthesis kit (Bio-Rad) on a 20 μL reaction as per manufacturer's instructions. RT-PCR was run on 10-μL volumes using iQ SYBR green Supermix (Bio-Rad) with 0.2 μM forward and reverse primers (Table 1) and 3 ng of cDNA. Amplification was done on a CFX96 TM IVD Real-Time PCR system (Bio-Rad) with a thermal cycle of 50 °C for 2 min, 95 °C for 2 min, and then 95 °C for 15 s and 60 °C for 30 s for a total of 40 cycles. Ct values of RT-PCR were normalized against the housekeeping gene (GAPDH) and analyzed using the $2^{\wedge}(-\Delta Ct)$ model to show relative gene expression.

**Total DNA and ATP, ALP and osteocalcin release quantification**. For DNA quantification, samples were harvested after 21 days of culture and subjected to 3 freeze-thaw cycles in liquid $N_2$. The ECM was digested for 16 h at 56 °C using a 50 mM Tris/1 mM EDTA/1 mM iodoacetamide solution containing 1 mg/mL Proteinase K. After digestion, samples were frozen-thawed again to facilitate DNA extraction. DNA was quantified using the CyQuant cell proliferation assay (ThermoFisher), following the manufacturer's instructions. ATP, ALP and osteocalcin release was measured from cell supernatants collected 30 min after sonication (24 h of culture). Quantification of ATP was carried out with an ATPlite luminescence assay system (PerkinElmer), following the manufacturer's instructions. ALP release and osteocalcin were measured via sandwich ELISA kits from LifeSpan BioSciences (LS-F3536) and R&D systems (DSTCN0), respectively, following the manufacturer's instructions.

**Statistical analysis and reproducibility**. Statistical analyses were performed using Graph Pad Prism 7.04. Sample size and significance are provided in the figure legends and information about statistical tests and raw data for the figures is included in the Source Data file. Biological experiments used at least three biological replicas, including microscopy images. Phase segregation was studied with five independent samples, including TEM and particle analysis. SEM and LSM images of phase segregation are representative of three replica. Images for particle analysis and fibronectin analysis were selected randomly. Printed scaffolds were also assigned randomly to different experimental groups. For biological experiments (gene expression analysis, DNA, ATP, ALP, and osteocalcin release) data were analyzed by regular two-way ANOVA without repeated measures or matching followed by Tukey's post hoc corrected for multiple comparisons test. Differences between conditions within an experimental group were tested using uncorrected Fisher's LSD test. For particle analysis a one-way ANOVA without matching or repeated measures, followed by Tukey's post hoc corrected for multiple comparisons test was used. α = 0.05 for all tests.

**Reporting summary**. Further information on research design is available in the Nature Research Reporting Summary linked to this article.

## Data availability

Data underlying the figures can be found in the figshare repository under the following url: https://doi.org/10.6084/m9.figshare.13537100, https://doi.org/10.6084/m9.figshare.13537067, https://doi.org/10.6084/m9.figshare.13537022, https://doi.org/10.6084/m9.figshare.13537007, https://doi.org/10.6084/m9.figshare.13536986, https://doi.org/10.6084/m9.figshare.13536977, https://doi.org/10.6084/m9.figshare.13536959. Source data are provided with this paper.

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

## Acknowledgements

The authors acknowledge the Texas A&M Health Science Center College of Medicine Institute for Regenerative Medicine at Scott & White who isolated and provided the cells through a grant from NCRR of the NIH (Grant #P40RR017447). The authors acknowledge the financial support from the European Commission under the ERC starting grant "Cell Hybridge" of the Horizon2020 framework program (Grant # 637308).

## Author contributions

S.C.-E. and L.M. conceived the idea. S.C.-E. prepared the samples, designed, and executed the experiments and analyzed the results. S.C.-E. and L.M. wrote the manuscript. L.M. supervised the project.

## Competing interests

The authors declare no competing interests.
