## [Peer Review File · Nature Communications]

Reviewers' Comments:

Reviewer #1:

Remarks to the Author:

This manuscript is about the 4D printing of a PCL/PLA blended scaffold using phase-segregation method. Authors' report the case in which the scaffold physically vibrates and stimulates stem cells when ultrasound is applied to it. The most crucial claim of the findings is, when ultrasonic waves are applied, the scaffold vibrates directly and mechanically expose the cells to vibrational stimuli. Also, the authors' main hypothesis is that it stimulates the voltage-calcium ion channel to promote bone differentiation. While the work is interesting, there are some major flaws that need to be addressed with the manuscript.

1. Further explanations and experiments are needed about the mechanism and effect of phase separation of PCL/PLA.

(1) A description of the reasons for phase separation and its advantages should be added.

(2) In Figure S3 and Figure S4, the result of PCL: PLA = 50:50 is omitted.

(3) The theoretical explanation about the phase-separated scaffold acting as a transducer is not clear.

(4) Since there is no analysis on the surface chemistry of Janus scaffolds, additional experimental data is required.

2. Overall, quantitative data needs to be included in this manuscript.

(1) Line 151-152 states that "Cells cultured in PLA and stimulated at 40 kHz had a higher proliferation rate, but their capability to deposit fibronectin was unaltered." As mentioned, this result needs to be verified to quantitatively analyze fluorescence images or by providing quantitative data.

(3) Quantitative data using MTT assay would be more appropriate instead of counting the number of cells in Figure 2C.

3. The reference supporting the "Line 197-198 that round HA being amorphous HA" seems to be insufficient. In addition, reference or experimental data is needed for the explanations why amorphous HA is more advantageous for bone formation than crystal-like HA.

4. The authors modeled the deflection of a single-layered scaffold. Since the scaffold used in the actual experiment has a three-dimensional shape, it is questionable whether there is a change in deflection according to the height difference, which might result in confusion for the readers. In addition, there seems to be a slight difference between the simulated data and the actual results. (between Figure 2B and Figure 3)

5. In the Introduction, it was mentioned that the technology could be used for non-invasive treatment, however, it is necessary to verify whether the actual ultrasonic stimulation is effectively transmitted to the implanted scaffold through in vivo model.

6. The authors argue that the Janus scaffold directly exerts a mechanical stimulus to the cell. For this argument to be persuasive, other factors that increase simultaneously compared to the control group must be additionally considered. The authors need to reinforce the data that can be further emphasized by decoupling the ultrasound-induced stimulation from mechanical stimulation.

7. Minor edit

(1) Line 99 "3,14 μm^2 (CI 95,3%) and 1,17 μm^2 (CI 96,01%)" Please check and revise

(2) Line 101 "1,3 (CI 95,1%) and 1,1 (CI 95,6%)" Please check and revise

(3) Line 161 -needs to be modified

(4) Line 20 polycaprolactone → polycaprolactone

(5) Line 169 and and → and

(6) Line 412 osteocalcin → osteocalcin

(7) Supporting data Figure S14. Ca²⁺ → Ca²⁺

Reviewer #2:

Remarks to the Author:

This manuscript proposed a 3D printed "Janus" scaffold that under ultrasound stimulation seems to favor bone generation. The method mixes PCL and PLA to create heterogeneous fillaments to be used to build the scaffold using a 3D FDM printer. The authors varied the PCL/PLA ratios to show that at 50:50 ratio, the mixture completely phase-segregate during the printing that involves melting and cooling creating one phase clearly adjacent to the other. The rest of the paper is characterizing the scaffold for its ultrasound response and cellular studies on them in the presence of ultrasound.

An array of standard methodologies---SEM, TEM, polarized and light scanning microscopy (LSM), were used for the characterization. PLA being a stiffer material than the PCL, expectedly the stiffness or elastic modulus of the "Janus" mixture was between them, and expectedly PLA showed the maximum deflection under nanoindentation and vibration amplitude for the same ultrasound excitation at (10,20,40 kHz).

hBMCs seeded in all three scaffolds with ultrasound excitation were compared for cell proliferation, fibronectin deposition. Osteogenic differentiation was studied in all three with different biomarkers. The pathway for enhanced osteogenic differentiation was hypothesized to be through a voltage gated Ca channel and verified by gene expression and blocking the channel.

The paper clearly states the methodology, and offers many-faceted detailed experimental studies with many results. It is interesting although some of the results are a bit spotty. I have also lessened enthusiasm due to the following issue

1. The claimed "Janus" character of the scaffold is a bit confusing. My own understanding is that nanoparticles with two sides having opposite characters---e.g. hydrophilic and hydrophobic---are called Janus. In contrast here, if I understood it clearly, one layer of PCL attached to a layer of PLA. It seems to be a sandwiched composite layer of material. The name "Janus" although catchy, I am not sure, is appropriate and could lead to confusion in the literature.

2. One major issue I have is the mechanical description of the response of the composite layer as a combination of a "damping layer" of PCL and an "active layer" of PLA. The PCL has a higher elastic modulus than PLA which explains its mechanical response. However, "damping" relates to resistive of viscous component. The authors didn't measure any viscous properties. The terms were used colloquially and would cause confusion.

3. Specifically, the whole paper is premised on such a composite structure of "active" and "damping" element may be the key to better cellular performance in Janus layer. To quote the paper

"The differences we observed in cell proliferation might be related to the pulse of the scaffold deflection, being the pulse width highest in PLA scaffolds, followed by Janus and PCL"

This statement is speculative and not based on any experiments that I can find. As stated before mixing two materials of different elasticities, one would expect a mechanical response of intermediate magnitude.

4. Why the above would give rise to better cellular response---cell proliferation in PLA and Janus are similar---in composite.

Here are specific places of the paper:

1. Line 104: "definite" instead of "define"

2. Line 119: what "computational modeling" not described.

It seems clear that the author found difference in different scaffold and under the particular situation, composite layer seems to perform better. However, it's not clear to me what caused this difference as opposed to "active" and "damping" description which fails me.

Reviewer #3:

Remarks to the Author:

Revision of the manuscript #: NCOMMS-2027251

Title: JANUS 3D printed dynamic scaffolds for nanodeflexion -driven bone regeneration

This work, which describes the physical and biological evaluations conducted on a new Janus 3D printed scaffold with properties of dynamic variation of its structure when subjected to an ultrasonic field, is particularly interesting for the modulation of the mesenchymal cell response in bone regeneration. The work is convincing but some methodological aspects are missing that would make the work more understandable. A major revision is needed.

Abstract and Introduction are well written and concise. Some misspellings. The last paragraph of the introduction from 'Our data... in my opinion should be carried over into the discussion. Furthermore, in the statement... enhanced differentiation correlated with increased formation and further... the term 'correlated' is not appropriate.

Results

The discussion section is missing but from the reading it emerges that it is one with the results. In my opinion it would be preferable to divide them as is the style of the journal. In addition, repeating some of the methodology in the results could be avoided by separating the information into their specific sections

Fabrication of 3D structures with...

What do the data mean (95.4% CI) or (95.9% CI) and so on. Do they refer to confidence intervals? Better to clarify.

Exploitation of Janus Scaffold...

The paragraph 'We seeded hBMSCs on the different scaffolds should be rewritten because it refers to figures 2c, 2d and 2e and talks about proliferation but in particular the graphs report cell density and cell proliferation normalized by scaffold deflection without defining whether the differences are significant.

Janus scaffold promote osteogenic differentiation...

From the bar plot it does not seem to me that osteocalcin is upregulated; the values of $2^{-\Delta\Delta Ct}$ are higher in US + than in US-, but still strongly lower than 1. Explain.

Methods

It would be necessary to add a figure showing the ultrasound stimulation and detection set-up as well as the measurement of scaffold deflection.

The primary BMSC cells (where human write hBMSCs in the text) used in the work are the same ones mentioned in the publication of the same authors (Stem Cells J 2020

<https://doi.org/10.1002/stem.3198>). In my opinion, it would be correct to quote the ethics committee that approved the study and the approval number and specify better what the aim of that study was.

Since numerous experiments have been carried out in the aforementioned work, as well as in this one, doubts arise in relation to the exact number of steps the cells have been used for, it is difficult to obtain many cells in P2. If necessary, provide details of the expansion procedure.

From rough calculations, as described in the materials and methods, it appears that approximately 135,000 cells per replicate were seeded for cell proliferation, matrix deposition and osteogenic differentiation tests. However, the number of replicates is not clear. In fact, in the Reporting summary on page 2 it is reported "Biological studies were performed 3 times with technical and biological triplicates all showing the same trend". So it would seem that they did 3 biological replicates for 3 types of tests and for each of these 3 technical replicates ($n = 3$ triplicates per test). The total of cells for seeding was to be 3.645×10^6 cells. Am I right?... To avoid misunderstandings, the authors should report the exact number of tests performed (n) specifying how many technical replicates were.

Why have the scaffolds been sterilized in ethanol which is not the normal sterilization procedure for implantable materials and it is well known that polymers undergo major changes with common sterilization methods (gamma, beta and ethylene oxide)? Doing biological studies on scaffolds whose post-sterilization structural stability is not certain is also somewhat of an end in itself. It is necessary to discuss this aspect in the discussion, perhaps as a limit.

Specify on which (matrix deposition?) And how many scaffolds the immunofluorescence was performed, as well as the evaluation of the expression analysis and of the Total DNA and ATP, ALP and osteocalcin quantification. As for, has gene expression been reported on the ddCT or the $2^{-\Delta\Delta Ct}$?

As for the statistical analysis paragraph, it should be better described, specifying the type of distribution of the data and their variance, so that the assumptions for conducting the ANOVA test are not violated. Also, many of the items flagged in the 'reporting summary' for Statistics are not defined in the text at all. I deduced from the results the factors involved in the ANOVA analysis and they are not defined in the statistical paragraph. There are no covariates. For the multiple comparison test was a correction for the p-values adopted? What α level was chosen to define the type I error, $p < 0.1$, $p < 0.05$ or $p < 0.01$? If $p < 0.01$ remove the significance of $p < 0.1$.

Figures

Pay attention to how the results are reported. Decimals must be indicated with the '.' and not with the ','.

Figures 3S and 4S. Better adapt the characters of the text of the axes (for the x axis write 20: 80,....). Note that in addition to the median, all data are reported as a beeswarm plot. I don't understand how 95% CI is displayed when not seeing specific error bars. When reporting the data with $p <$, the values 0.0005, 0.005, 0.05 or 0.0001, 0.001, 0.01 are used by convention; it doesn't make much sense to write $p < 0.002$ or $p < 0.0021$ etc. because then we write directly $p = 0.0021$. In S3 and S4 the *** $p < 0.0002$ is not in the graphs. Also there must be an error in the values of p which are equal for the two figures.

Figures 2, S12 and S13 are missing the specifications of the asterisks values.

Tables.

In table 1 being a list of human genes, in the first column the genes should be written in uppercase and italics

Table S1 and S2 do not report the number of tests and for S1 and S3 it is not clear whether they are means without standard deviations or single data.

Reviewer #1 (Remarks to the Author):

This manuscript is about the 4D printing of a PCL/PLA blended scaffold using phase-segregation method. Authors' report the case in which the scaffold physically vibrates and stimulates stem cells when ultrasound is applied to it. The most crucial claim of the findings is, when ultrasonic waves are applied, the scaffold vibrates directly and mechanically expose the cells to vibrational stimuli. Also, the authors' main hypothesis is that it stimulates the voltage-calcium ion channel to promote bone differentiation. While the work is interesting, there are some major flaws that need to be addressed with the manuscript.

We acknowledge the reviewer for the overall positive evaluation of the manuscript.

1. Further explanations and experiments are needed about the mechanism and effect of phase separation of PCL/PLA.

We acknowledge the reviewer for his/her curiosity on these phase segregating systems. We find that they are indeed very interesting. Please, find our detailed answers below.

(1) A description of the reasons for phase separation and its advantages should be added.

The mechanism of phase segregation of PCL and PLA blends has been extensively described on manuscripts dedicated to them. Although the miscibility parameters of PCL ($(9.2 \text{ cal/cm}^3)^{1/2}$) and PLA ($(10.1 \text{ cal/cm}^3)^{1/2}$) are relatively similar (*Polymer* **1990**, 31 (7), 1187), there are no favorable interactions between them, which renders them immiscible. We have included a clarification on the text that now reads:

Page 3-4, line 26-2: "Despite accounting for a similar Hildebrand solubility parameters (PCL $9.2 \text{ (cal/cm}^3)^{1/2}$ and PLA $10.1 \text{ (cal/cm}^3)^{1/2}$)³⁰ PCL and PLA have no favorable interactions and are therefore immiscible. Phase segregation of this polymer system has extensively been studied due to the favorable reduction of brittleness of PLA when blended with PCL^{31, 32, 33}. Thus, initial blending to form filaments with PLA:PCL ratios..."

(2) In Figure S3 and Figure S4, the result of PCL: PLA = 50:50 is omitted.

We apologize for the confusion this might have generated. The measurements shown are a particle analysis of the phase segregation from TEM images. That is, the area of each individual formed PLA or PCL domain within the matrix. To calculate the area on the 50:50 sections by TEM we would need to have a full overview of the section itself, otherwise the values would be restricted to the imaged area. A full section occupies a large area, out of the boundaries of the grid structure itself, meaning that this one is visible and makes the measurements not possible (unless digitally removed). The sections of the Janus blends are particularly difficult to be microtomed as these two polymers account for very different mechanical properties, which is reflected on the wrinkles presented on the sections when deposited on the TEM grids. This wrinkling, further introduces an error into the measurement of the total area of each phase.

Thus, we have instead included a measurement of the % of phase segregated area of PLA within Janus scaffolds calculated from SEM images and added it as a separate figure. This is now Figure S6.

"Figure S6. Analysis of the % of the total cross-sectional area occupied by the PLA phase on scaffold fibers for a PLA:PCL ratio of 50:50. Bars show median and 95% CI. n = 10"

We have also modified the text to cover this extra data that reads:

Page 4, line 20-22: "The phase segregation on Janus fiber`s cross-sections corresponded to the ratio of the two polymers on the blend, with an average occupied area of PLA phase of 48.2 % \pm 8.2 (Figure S6)".

We have corrected the caption of the figure S3 and S4 that now read:

“Figure S3. Analysis of the area (a) and aspect ratio (b) of the phase segregated particles on the cross-sectional direction for PLA:PCL ratios of 20:80, 30:70, 40:60, 60:40, 70:30 and 80:20. Bars show median and 95% CI. Statistical significance was calculated from 1-way ANOVA. (****) $p < 0.0001$, (***) $p < 0.001$, (**) $p < 0.01$ and (*) $p < 0.1$. $n = 5$.”

“Figure S4. Analysis of the area (a) and aspect ratio (b) of the phase segregated particles on the longitudinal direction of the scaffold fibers for PLA:PCL ratios of 20:80, 30:70, 40:60, 60:40, 70:30 and 80:20. Bars show median and 95% CI. Statistical significance was calculated from 1-way ANOVA. (****) $p < 0.0001$, (***) $p < 0.001$, (**) $p < 0.01$ and (*) $p < 0.1$. $n = 5$ ”

(3) The theoretical explanation about the phase-separated scaffold acting as a transducer is not clear. We agree with the reviewer that further context would be needed. We have included an explanation of this concept and measurements that support our hypothesis.

Page 7, line 7-16: “Ultrasound transducers are fabricated as a combination of a damping material (dissipating the energy and remaining static or with very little vibration) that is placed underneath the active or vibrating material (storing energy). The combinatorial response of the sandwich composite results on a reduced pulse length or width (and amplitude) of the transmitted wave (Figure 3, schematic). Similarly, Janus scaffolds account for a PCL phase with a higher energy dissipation or damping potential (tangent δ at 25 °C and 0.1Hz of $11.3 \cdot 10^{-2} \pm 0.4 \cdot 10^{-2}$) than the PLA phase (tangent δ at 25 °C and 0.1 of $6.3 \cdot 10^{-2} \pm 0.9 \cdot 10^{-2}$) that presents a higher energy storage potential, with a storage and loss moduli of 42.8 ± 5.9 MPa and 4.9 ± 0.5 MPa for PCL and 81.6 ± 13.7 MPa and 5.4 ± 1.1 MPa for PLA (Figure S14 and S15). The resulting Janus account for a storage modulus of 89.5 ± 12.2 MPa, a loss modulus of 6.8 ± 0.3 MPa and a tangent δ of $7.6 \cdot 10^{-2} \pm 0.4 \cdot 10^{-2}$.”

And,

Page 8, line 1-3: “The combination of these two materials, PLA as active and PCL as damping material, in Janus scaffolds resulted in a reduced amplitude of 36 ± 4 nm and pulse width (1 cycle, 0.1 s), and an increased PRF of 2.17 Hz as compared to PLA scaffolds, as it occurs on ultrasound transducers.”

(4) Since there is no analysis on the surface chemistry of Janus scaffolds, additional experimental data is required. We acknowledge the detailed revision and interest of the reviewer. The surface of the fibers is composed of PLA and PCL distributed on the upper and bottom areas of the fibers, forming the Janus. We considered to perform a characterization of the surface chemistry of the material as a further proof of the phase segregation but the minimum differences on the composition of the two polymers difficult the task. Typical surface chemistry characterization techniques such as TOF-MALDI or X-ray photoelectron spectroscopy provide local information of the composition and the differentiation between the chemical structure of PLA and PCL is not revealed, as the binding energies of the elements are virtually the same. Thus, we decided to focus on the presented techniques (TEM, SEM, POM and LSM) that provide us information of the structure that can be related to the composition.

We would like to point out that biological experiments were produce on fibronectin coated scaffolds to avoid any differences arising from different surface chemistries.

2. Overall, quantitative data needs to be included in this manuscript.

We have had 27 graphs and 3 tables on the initial version of the manuscript that show quantitative data, and believe that this is quite a significant number of it for a “communication” type manuscript. On top of these, we have presented computational modelling and imaging. After reviewing the manuscript, we have now 34 graphs and 3 tables.

(1) Line 151-152 states that “Cells cultured in PLA and stimulated at 40 kHz had a higher proliferation rate, but their capability to deposit fibronectin was unaltered.” As mentioned, this result

needs to be verified to quantitatively analyze fluorescence images or by providing quantitative data. We have introduced a quantification of the area covered with fibronectin (%) on the scaffolds calculated from the fluorescence images presented in Figure 2, f. This is now Figure S12:

*“Figure S12. Quantification of the area covered with fibronectin on ultrasound stimulated (+US) and non-stimulated Janus, PCL and PLA scaffolds (-US) after 2 weeks of BMSC culture. The data is shown as mean \pm standard deviation. Statistical significance was calculated by two-way ANOVA with Tukey’s multiple comparison test; (****) $p < 0.0001$, (***) $p < 0.001$, (**) $p < 0.01$ and (*) $p < 0.1$ $n = 3$.”*

We have also included this measurement on the methods section:

Page 16, line 21-22: “Quantification of the area covered by fibronectin was done using Fiji free software.”

(3) Quantitative data using MTT assay would be more appropriate instead of counting the number of cells in Figure 2C.

We agree with the reviewer. Generally, assays such as MTT or direct quantification of the total DNA are more accurate than cell counting. We have included measurements of the quantity of total DNA on figures 4 and 5 as these are long-term cultures (3 weeks) with a high amount of cells. In figure 2c, we chose to count nucleus from the scaffolds as this ensured that the quantified cells are solely those that are directly on top of the scaffold and therefore subjected to mechanical deflection. Using MTT or DNA quantification assays could introduce errors on the results arising from cells growing in between the scaffolds fibers. Thus, we believe that counting cell nucleus on this particular case, would give a more accurate result.

3. The reference supporting the "Line 197-198 that round HA being amorphous HA" seems to be insufficient. In addition, reference or experimental data is needed for the explanations why amorphous HA is more advantageous for bone formation than crystal-like HA.

We acknowledge the reviewer from bringing this to our attention.

Amorphous HA is not more advantageous than crystalline HA. Yet, it is better than simple Ca^{2+} salts.

Page 9, line 2: *“However, a detailed observation of the deposited minerals by SEM revealed that BMSCs cultured on PCL underwent what has been defined as “chemical differentiation”, depositing minerals with a smooth and big crystal-like morphology characteristic of Ca^{2+} salts³⁵. Minerals deposited on Janus scaffolds, however, showed a rounded and porous morphology characteristic of amorphous hydroxyapatite^{35, 36}”*

Hydroxyapatite in bone is present as a crystalline form. Amorphous calcium phosphates transform onto crystalline hydroxyapatite as they exit the cell. It transforms from a spherulite or globular shape onto elongated platelet-like morphology of native crystalline HA (PNAS 2010, 107 (14), 6316.; Nature Communications 2018, 9 (1), 4170; Chemical reviews 2008, 108 (11), 4754).

The morphology associated to amorphous and crystalline forms of HA are well known and have been reported extensively (*Nature Communications* 2018, 9 (1), 4170); *ACS Nano* 2016, 10 (7), 6826.; *PNAS* 2010, 107 (14), 6316; *PNAS* 2008, 105 (35), 12748.)

We have further clarified this on the text that now reads:

Page 9, lines 5-8: "Minerals deposited on Janus scaffolds, however, showed a rounded and porous morphology characteristic of amorphous hydroxyapatite or calcium phosphate^{35, 36}. During bone mineralization, amorphous and globular shaped calcium phosphates undergo mineralization to form carbonated hydroxyapatite, thus indicating the bone formation potential on Janus scaffolds^{37, 38, 39,}

40 "

4. The authors modeled the deflection of a single-layered scaffold. Since the scaffold used in the actual experiment has a three-dimensional shape, it is questionable whether there is a change in deflection according to the height difference, which might result in confusion for the readers. In addition, there seems to be a slight difference between the simulated data and the actual results. (between Figure 2B and Figure 3)

We acknowledge the reviewer for bringing this into our attention. The modelled and used 3D printed objects have the same structure and dimensions (height, length, diameter of the fibers and number of layers). We have clarified this fact in the text.

Page 6, line 1-3: "We seeded hBMSCs on the different scaffolds and cultured them for 7 days applying 30 minutes of stimulation daily at either 0, 10, 20 or 40 kHz using the same structures as modelled in figure 2b (Figure 2c, d, e and Figure S11)."

There is indeed a difference between the simulated data and the actual measurements. We have mentioned this on the manuscript and pointed out that these two followed the same trend (page 7, line 15-17). The difference from measured and modelled data can arise from multiple factors. The model is an ideal system with no external vibrations or reflection that can result. The echoed wave can then sum to the incident wave, resulting on an incident wave of greater amplitude. An incident wave of greater amplitude will then result on a greater deflection. We have clarified this on the text.

Page 8, line 5-7: "The difference on the absolute values of the deflection probably arise from the result (sum) of the incident and echoing waves not contemplated in the model, resulting on greater deflection amplitudes."

5. In the Introduction, it was mentioned that the technology could be used for non-invasive treatment, however, it is necessary to verify whether the actual ultrasonic stimulation is effectively transmitted to the implanted scaffold through in vivo model.

We agree with the reviewer that presentation of an in-vivo model would be interesting to this end. We have planned an in-vivo experiment that will follow up to this communication. On this communication we aim to show (i) the novel method to create 3D printed objects with controlled chemistries via phase segregation, (ii) the concept of using ultrasounds as trigger to stimulate 3D printed objects and (iii) their potential exploitation for the regeneration of bone.

However, there is already evidence showing the capability to transmit ultrasound waves to cells on in-vivo situations, as indicated in the introduction. We have stated this need of further investigation in the manuscript, page 11, that now reads:

Page 11, line 13-16: "Taken altogether, we propose that remote activation of Janus scaffolds could present an ideal alternative to traditional "static" implants, providing "on-command" stimulation of cells. However, their applicability on in-vivo situations is yet to be investigated."

6. The authors argue that the Janus scaffold directly exerts a mechanical stimulus to the cell. For this argument to be persuasive, other factors that increase simultaneously compared to the control group must be additionally considered. The authors need to reinforce the data that can be further emphasized by decoupling the ultrasound-induced stimulation from mechanical stimulation.

We agree with the reviewer: being able to decouple ultrasound stimulation from mechanical stimulation would be ideal. However, (ultra)sound waves are pressure waves. This implies a

pressure or force being applied. To be able to decouple these two stimulations, one would need to introduce a fourth material, one that does not vibrate.

For the particular conditions studied on this manuscript, we would need to find a material that is biocompatible and with a significantly lower mechanical properties and equal or higher density than the presented materials. Alternatively, a biocompatible material with significantly higher density. Satisfying these two conditions is complicated. Another possibility would be the use of simple cell pellet culture. However, here again, the pressure (sound) waves would also introduce a mechanical stimulation. Thus, we used two controls here. (i) PCL alone as a low vibrating material, and (ii) PLA as vibrating material.

7. Minor edit

We acknowledge the reviewer for the careful correction of the manuscript. We have addressed all the minor edits.

(1) Line 99 "3,14 μm^2 (CI 95,3%) and 1,17 μm^2 (CI 96,01%)" Please check and revise

(2) Line 101 "1,3 (CI 95,1%) and 1,1 (CI 95,6%)" Please check and revise

We have modified the text, substituting "," for "."

(3) Line 161 -needs to be modified

We have modified the text that now reads: "*Applying a Fast Fourier Transform (FFT) to the recorded data, the response was decoupled onto a wave with the characteristic frequency of the input signal (approximately 40 kHz) and a secondary wave (the pulse) that was dependent on the material used.*"

(4) Line 20 polycaprolactone → polycaprolactone

(5) Line 169 and and → and

(6) Line 412 osteocalcin → osteocalcin

(7) Supporting data Figure S14. Ca^{+2} → Ca^{2+}

We acknowledge the reviewer for the careful evaluation. We have corrected all these mistakes.

Reviewer #2 (Remarks to the Author):

This manuscript proposed a 3D printed "Janus" scaffold that under ultrasound stimulation seems to favor bone generation. The method mixes PCL and PLA to create heterogeneous filaments to be used to build the scaffold using a 3D FDM printer. The authors varied the PCL/PLA ratios to show that at 50:50 ratio, the mixture completely phase-segregate during the printing that involves melting and cooling creating one phase clearly adjacent to the other. The rest of the paper is characterizing the scaffold for its ultrasound response and cellular studies on them in the presence of ultrasound.

An array of standard methodologies---SEM, TEM, polarized and light scanning microscopy (LSM), were used for the characterization. PLA being a stiffer material than the PCL, expectedly the stiffness or elastic modulus of the "Janus" mixture was between them, and expectedly PLA showed the maximum deflection under nanoindentation and vibration amplitude for the same ultrasound excitation at (10,20,40 kHz).

hBMCs seeded in all three scaffolds with ultrasound excitation were compared for cell proliferation, fibronectin deposition. Osteogenic differentiation was studied in all three with different biomarkers. The pathway for enhanced osteogenic differentiation was hypothesized to be through a voltage gated Ca channel and verified by gene expression and blocking the channel. The paper clearly states the methodology, and offers many-faceted detailed experimental studies with many results. It is interesting although some of the results are a bit spotty. I have also lessened enthusiasm due to the following issue

We acknowledge the reviewer for the overall positive evaluation of the manuscript.

1. The claimed "Janus" character of the scaffold is a bit confusing. My own understanding is that nanoparticles with two sides having opposite characters---e.g. hydrophilic and hydrophobic---are called Janus. In contrast here, if I understood it clearly, one layer of PCL attached to a layer of PLA.

It seems to be a sandwiched composite layer of material. The name "Janus" although catchy, I am not sure, is appropriate and could lead to confusion in the literature.

We agree with the reviewer on the definition of Janus; it is a structure composed of two materials with "opposing" characteristics. However, the term is not exclusive to nanoparticles. It has been extensively used to define thin films, membranes, cylinders and ribbons (*Angew. Chem. Int. Ed.* 55, 13398–13407 (2016); *Angew. Chem. Int. Ed.* 2014, 53 (22), 5524.; *Nature materials* 2006, 5 (5), 365.; *Macromolecules* 2011, 44 (23), 9221).

Moreover, this term has also recently been applied to 3D printed fibers using co-extrusion (*ACS Applied Materials & Interfaces* 2017, 9 (5), 4873).

In our particular case, one of the materials is amorphous and active (PLA) and the second is crystalline and damping (PCL), as we have shown on the article and supporting information (Figure 3, Figure S2, S14 and S15). The PCL and PLA phase are not attached to each other (were a sandwich definition would be better suited), but arise from a polymer blend and further phase segregate during processing. Thus, we believe the use of the world Janus to be well suited.

2. One major issue I have is the mechanical description of the response of the composite layer as a combination of a "damping layer" of PCL and an "active layer" of PLA. The PCL has a higher elastic modulus than PLA which explains its mechanical response. However, "damping" relates to resistive of viscous component. The authors didn't measure any viscous properties. The terms were used colloquially and would cause confusion.

We agree with the review that damping is ascribed to the viscoelastic character of the material and also related to microstructural properties. That is dislocations, phase boundaries and other defects that slip upon the vibrational force, thus dissipating the energy (*J. Mater. Sci.* 2001, 36 (24), 5733.)

In our case, both the flexural modulus and the density of PLA are higher (as already described in the manuscript), hence behaving as the active layer.

To further strengthen this very important point that the reviewer raised, and to demonstrate the damping character of PCL we have performed dynamic mechanical thermal analysis. We now report the storage modulus, loss modulus and $\tan\delta$ of the materials (PLA, PCL and Janus) measured at 0.1Hz with a temperature ramp and, at 37C and a frequency sweep. The $\tan\delta$ is in both cases higher for PCL, followed by Janus and lower for PLA, which demonstrates its damping character.

We have clarified this on the text.

Page 7, line 7-16: "Ultrasound transducers are fabricated as a combination of a damping material (dissipating the energy and remaining static or with very little vibration) that is placed underneath the active or vibrating material (storing energy). The combinatorial response of the sandwich composite results on a reduced pulse length or width (and amplitude) of the transmitted wave (Figure 3, schematic). Similarly, Janus scaffolds account for a PCL phase with a higher energy dissipation or damping potential (tangent δ at 25 °C and 0.1Hz of $11.3 \cdot 10^{-2} \pm 0.4 \cdot 10^{-2}$) than the PLA phase (tangent δ at 25 °C and 0.1 of $6.3 \cdot 10^{-2} \pm 0.9 \cdot 10^{-2}$) that presents a higher energy storage potential, with a storage and loss moduli of 42.8 ± 5.9 MPa and 4.9 ± 0.5 MPa for PCL and 81.6 ± 13.7 MPa and 5.4 ± 1.1 MPa for PLA (Figure S14 and S15). The resulting Janus account for a storage modulus of 89.5 ± 12.2 MPa, a loss modulus of 6.8 ± 0.3 MPa and a tangent δ of $7.6 \cdot 10^{-2} 0.4 \cdot 10^{-2}$."

We have also included the corresponding figures on the supporting information (Figure S14 and S15) and referenced to them on the text.

"Figure S14. Representative dynamic mechanical thermal analysis curves showing the (a) storage (G') and loss (G'') moduli and (b) the resulting tangent δ (\tan) of PLA, PCL and Janus 3D printed fibers at a frequency of 0.1 Hz."

"Figure S15. Representative dynamic mechanical analysis curves showing the (a) storage (G') and loss (G'') moduli and (b) the resulting tangent δ (\tan) of PLA, PCL and Janus 3D printed fibers at 37 °C and over a one decade frequency sweep."

And, a description of mechanical analysis on the methods section

Page 18, line 5-17:

"Mechanical properties of the materials

The flexural modulus of PCL, PLA and 50:50 blends was measured under 3-point bending from extruded filaments of 2 mm diameter mounted on a 1 cm support span. A TA ElectroForce (TA Instruments) mechanical tester equipped with a 45 N load cell and controlled with Wint7 software was used to deform the samples at a strain rate of 0.01 mm/s. The experiments were run until creep (or until the maximum applicable load was reached). The flexural modulus was calculated from the slope of the linear region of recorded force-displacement curves. The storage and loss moduli and the tangent δ of PCL, PLA and Janus scaffolds were measured from 3D printed single fibers of 400 μm diameter using a TA Q800 dynamic mechanical thermal analyzer. Experiments were run with a thermal sweep from 20 – 65 °C, a 3 °C temperature ramp, a dynamic strain of 0.5 % and fixed frequency of 0.1 Hz from which the storage modulus, loss modulus and tangent δ are reported as mean \pm SD, $n = 3$. Experiments were also run under a frequency sweep of 0.1 – 100 Hz at a fixed temperature of 37 °C using a strain of 0.5 %. All PLA samples broke above 1.6 Hz. $n = 3$ "

3. Specifically, the whole paper is premised on such a composite structure of "active" and "damping" element may be the key to better cellular performance in Janus layer. To quote the paper "The differences we observed in cell proliferation might be related to the pulse of the scaffold deflection, being the pulse width highest in PLA scaffolds, followed by Janus and PCL" This statement is speculative and not based on any experiments that I can find. As stated before mixing two materials of different elasticities, one would expect a mechanical response of intermediate magnitude.

We agree with the reviewer, the manuscript is based on the concept of an active and a damping material. We have shown the damping character of PCL with measurements of the loss and storage modulus and the resulting $\tan \delta$ (see above). We have also shown that indeed there is a difference on the pulse on the scaffold deflection in Figure 3 and discussed it on page 7, indicating the difference on pulse amplitude, width and repetition frequency that matched that one for an active and damping materials. We ascribed this difference to the pulse width, as it has previously been shown to affect cell proliferation (page 2 of the manuscript, reference 18). We further tried to indirectly show the effect of the pulse on long-term experiments of osteogenesis and showing the actual activation of calcium ion channels. Thus, we believe that our hypothesis is now well supported.

4. Why the above would give rise to better cellular response---cell proliferation in PLA and Janus are similar---in composite.

We agree with the reviewer, this is a very interesting effect. Multiple studies showed in-vivo the preferred use and better results of pulsed over continuous ultrasounds for the treatment of musculoskeletal disorders (*Osteoarthritis and Cartilage* 2014, 22 (8), 1090.; *Int J Clin Exp Pathol* 2014, 7 (2), 779.). Moreover, as we discussed on the introduction, it has been shown that the pulse of the ultrasound wave has an impact on cell response (page 2, line 13). However, the exact reasons for this response are yet not known. Thus, our Janus scaffolds would also serve as platform to investigate these effects on a follow up study.

Here are specific places of the paper:

We acknowledge the reviewer for the detailed revision of the manuscript. We have addressed these comments on the text.

1. Line 104: "definite" instead of "define"

We have corrected this mistake on the text.

2. Line 119: what "computational modeling" not described.

We have clarified this on the text that now reads: ". Therefore, the flexural modulus of the materials was measured under 3-point bending and used for the computational modelling of the scaffold deflection under ultrasound stimulus."

It seems clear that the author found difference in different scaffold and under the particular situation, composite layer seems to perform better. However, it's not clear to me what caused this difference as opposed to "active" and "damping" description which fails me.

We agree with the reviewer that this is an intriguing effect.

We have now demonstrated with additional data the "damping" character of PCL and active character of PLA as a measurement of the storage modulus, loss modulus and tangent δ (see above).

We hypothesized, as discussed above, that the difference on cellular behavior is a result of the pulsed stimulation imparted by the Janus scaffolds. We also showed that this effect appears to be related to the activation of calcium ion channels. To further decipher the exact mechanism (signaling pathway) by which a pulsed excitation is better, a second study would be needed after this initial proof of concept where we also describe the fabrication process and the response of the materials.

Reviewer #3 (Remarks to the Author):

Revision of the manuscript #: NCOMMS-2027251

Title: JANUS 3D printed dynamic scaffolds for nanodeformation -driven bone regeneration

This work, which describes the physical and biological evaluations conducted on a new Janus 3D printed scaffold with properties of dynamic variation of its structure when subjected to an ultrasonic field, is particularly interesting for the modulation of the mesenchymal cell response in bone regeneration. The work is convincing but some methodological aspects are missing that would make the work more understandable. A major revision is needed.

We greatly acknowledge the reviewer for the positive evaluation of the manuscript.

Abstract and Introduction are well written and concise. Some misspellings. The last paragraph of the introduction from 'Our data... in my opinion should be carried over into the discussion. Furthermore, in the statement... enhanced differentiation correlated with increased formation and further... the term 'correlated' is not appropriate.

We acknowledge the reviewer for the positive evaluation. We have amended some parts of the text.

Page 2 line 16: "*Some examples of low frequency ultrasound stimulation of cell substrates have shown the relevance of these to induce osteogenic differentiation of cells but, this concept has not been exploited as functional 3D scaffolds, neither had the possibility of extrapolation to realistic in-vivo situation*^{22, 23}."

Page 3, line 6: "We" instead of "we".

Page 3, line 16: "*connected to*" instead of "*correlated with*".

We have retained the last paragraph of the introduction as we believe it helps the reader to understand already in the introduction the impact of the manuscript but we have included, as suggested by the reviewer, a discussion section at the end of the manuscript.

Results

The discussion section is missing but from the reading it emerges that it is one with the results. In my opinion it would be preferable to divide them as is the style of the journal. In addition, repeating some of the methodology in the results could be avoided by separating the information into their specific sections.

We have divided the text after the result section to conform a new discussion section, as the reviewer suggested and included some new sentences that bring perspective to the section. However, we have kept the text within the results section. Given the multidisciplinary character of the work reported, we believe that certain explanation and guidance throughout the manuscript would be preferable.

Page 10, line 14 – page 11, line 17:

"Discussion

In summary, we presented two new alternative routes to 4D printing: in-situ phase segregation to control spatially the composition of the printed structure, and ultrasound stimulation to remotely activate the deflection of the scaffolds. Varying the ratio of the polymer blend allowed controlling the phase formation to give rise to particles, ellipsoidal phases or Janus structures that were spontaneously formed during the printing process. The formation of such phase-segregated structures provides additional control for 3D printing strategies, developing the technology towards 4D printing. Particularly, the control over the chemistry could be exploited for the selective functionalization of the different formed phases and thus further control the cell phenotype. Although 4D printing has been previously shown, the reported systems are so far either non-reversible (1-way morphological changes) or non-compatible with biological systems.^{25, 26, 27, 28, 46} Thus, we believe this to be the first report of truly reversible dynamic scaffolds.

Theoretical models proved the use of the scaffolds feasible as dynamic implants that could be externally activated "on-command". PCL and PLA scaffolds responded to ultrasound as damping and deflecting materials, respectively. Combinations of these as Janus structures resulted in shorter pulse widths and smaller deflection. These properties of the scaffolds directly affected cell proliferation, matrix deposition and osteogenic differentiation of cultured hBMSCs as previously reported by others when ultrasound stimulation is directly applied to cells or on iv-vivo situations^{18, 47, 48}. Ultrasound stimulation of PCL scaffolds showed no significant effect in cell proliferation or matrix deposition, but a small increase in gene expression of some osteogenic markers and the deposition of Ca²⁺ salts, ascribed to a chemical differentiation.²⁰ Stimulation of PLA scaffolds resulted on an increased cell proliferation, but no effect on cell differentiation. Stimulation of Janus scaffolds affected hBMSCs with increased cell proliferation and expression and deposition of osteogenic markers.

Activation of L-VGCC of hBMSCs has been suggested to be key during osteogenic differentiation of hBMSCs^{43, 44}. Here, we showed that L-VGCC were activated on cells cultured in

stimulated Janus scaffolds, and that blocking these ion channels cancelled the cellular effect of ultrasound stimulation. This result is in accordance with previous studies that revealed the involvement of ion channels during mechanical, electrical and sonic stimulation of various types of stem cells^{49, 50, 51, 52, 53}. Taken altogether, we propose that remote activation of Janus scaffolds could present an ideal alternative to traditional "static" implants, providing "on-command" stimulation of cells. However, their applicability on in-vivo situations is yet to be investigated. Moreover, common sterilization processes such as ethylene oxide or gamma irradiation would need to be tested for their potential structural damage to the polyesters used herein."

Fabrication of 3D structures with...

What do the data mean (95.4% CI) or (95.9% CI) and so on. Do they refer to confidence intervals? Better to clarify.

We thank the reviewer for the detailed revision of the manuscript. Yes, this represents confidence intervals. We have included this clarification on the text. Page4, line 12: "(*Confidence Interval (CI) 95,4%*)"

Exploitaion of Janus Scaffold...

The paragraph 'We seeded hBMSCs on the different scaffolds should be rewritten because it refers to figures 2c, 2d and 2e and talks about proliferation but in particular the graphs report cell density and cell proliferation normalized by scaffold deflection without defining whether the differences are significant.

We thank the reviewer to bring this into our attention.

The normalized cell proliferation plotted in figure 2d is indeed the number of cells stimulated at a given frequency / cells with no stimulation. We have corrected the text to clarify this misleading term. We have defined it on the beginning of the paragraph as "*proliferation relative to non-stimulated cultures*" and carried on with the term "*normalized proliferation*" as indicated on the figure.

The data points are distributed along different deflection (X) values and hence, showing significance between "groups" is not possible (there are no groups). To properly show the difference between groups, we have included a new graph Figure S11 showing the normalized proliferation in function of the stimulation frequency.

Figure S11. *hBMSC number at 10, 20 or 40 kHz stimulation as normalized to the cell number at 0 kHz (normalized cell proliferation, %) after 7 days of culture on PLA, Janus and PCL scaffolds. Statistical significance was calculated by two-way ANOVA with Tukey's multiple comparison test; (****) $p < 0.0001$, (***) $p < 0.001$, (**) $p < 0.01$ and (*) $p < 0.1$. $n = 3$."*

Janus scaffold promote osteogenic differentiation...

From the bar plot it does not seem to me that osteocalcin is upregulated; the values of 2^{-ddCt} are higher in US+ than in US-, but still strongly lower than 1. Explain.

On these data we are plotting "relative expression" that is, 2^{-ddCt} ". The absolute values are lower than 1 because the housekeeping gene, GAPDH in this case, has a higher expression. This is expected as GAPDH is highly expressed on cells and tissues. We chose to use relative expression instead of 2^{-ddCt} (fold increase) as some genes were not expressed or not detectable in "untreated" hBMSC samples (before differentiation), such as CACNA1. Moreover, using non-stimulated (0KHz) samples as controls to plot of 2^{-ddCt} would have been possible. But then, the real values behind this calculation would be "hidden" and can become misleading.

We have clarified this on the methods section.

Page 20, line 19-20: "Ct values of RT-PCR were normalized against the housekeeping gene (GAPDH) and analyzed using the $2^{-\Delta Ct}$ model to show relative gene expression."

Methods

It would be necessary to add a figure showing the ultrasound stimulation and detection set-up as well as the measurement of scaffold deflection.

We have further clarified the used set-ups as suggested by the reviewer, including a Figure in the supporting info Figure S7. We have referenced this figure on the main text as well as methods section.

"Figure S7. Schematic representation of the ultrasound (A) detection set-up, measurement of deflection (B) and cell (C) stimulation."

The primary BMSC cells (where human write hBMSCs in the text) used in the work are the same ones mentioned in the publication of the same authors (Stem Cells J 2020 <https://doi.org/10.1002/stem.3198>). In my opinion, it would be correct to quote the ethics committee that approved the study and the approval number and specify better what the aim of that study was.

Since numerous experiments have been carried out in the aforementioned work, as well as in this one, doubts arise in relation to the exact number of steps the cells have been used for, it is difficult to obtain many cells in P2. If necessary, provide details of the expansion procedure. We thank the reviewer for notifying this lack of information. We have included a reference where the initial cell harvest and culture is described. We have also added to the methods subsection "cell culture" the further clarifications.

"Cryopreserved vials at passage 2 were plated at a density of 1,000 cells/cm² and cultured in basal media consisting of alpha-MEM media supplemented with Glutamax (Gibco) and 10% fetal bovine

serum (FBS) (Sigma-Aldrich). Cells were subcultured at 80 % confluence. All experiments were performed at cell passage 5.

Details of the grant approval are provided on the acknowledgement section that reads:

“The authors acknowledge the Texas A&M Health Science Center College of Medicine Institute for Regenerative Medicine at Scott & White who isolated and provided the cells through a grant from NCRR of the NIH (Grant #P40RR017447).”

From rough calculations, as described in the materials and methods, it appears that approximately 135,000 cells per replicate were seeded for cell proliferation, matrix deposition and osteogenic differentiation tests. However, the number of replicates is not clear. In fact, in the Reporting summary on page 2 it is reported "Biological studies were performed 3 times with technical and biological triplicates all showing the same trend". So it would seem that they did 3 biological replicates for 3 types of tests and for each of these 3 technical replicates (n = 3 triplicates per test). The total of cells for seeding was to be 3.645×10^6 cells. Am I right?... To avoid misunderstandings, the authors should report the exact number of tests performed (n) specifying how many technical replicates were.

Each of the scaffolds was seeded with 45,000 cells. 15,000 cell/cm², samples were 3cm². Each experiment contained 3 triplicates. We run the experiments 3 times, n = 3. From each sample, the measurements were read 3 times. We have included this on the methods section.

“All the experiments and analysis were performed in triplicates, n = 3.”

The types of tests were many more. PCR, immunofluorescence, ELISAs, SEM, histology, DNA, etc. The total of cells for seeding depended on the experiment. For example, proliferation only included 2 conditions (+US and -US), 3 material (PCL, PLA and Janus) and 3 triplicates. However, differentiations contained the 2 conditions (+US, -US), the 3 material (PCL, PLA and Janus) and all multiplied by 2, NFP blocking and not blocked. Then the amount of tests where 6 x 3 replica. These would be 216 samples, at 45k cell per sample, 9,72 mill cells.

Why have the scaffolds been sterilized in ethanol which is not the normal sterilization procedure for implantable materials and it is well known that polymers undergo major changes with common sterilization methods (gamma, beta and ethylene oxide)? Doing biological studies on scaffolds whose post-sterilization structural stability is not certain is also somewhat of an end in itself. It is necessary to discuss this aspect in the discussion, perhaps as a limit.

We have included this on the discussion section that reads.

Page 11, line 15: “...However, their applicability on in-vivo situations is yet to be investigated. Moreover, common sterilization processes such as ethylene oxide or gamma irradiation would need to be tested for their potential structural damage to the polyesters used herein.”

Specify on which (matrix deposition?) And how many scaffolds the immunofluorescence was performed, as well as the evaluation of the expression analysis and of the Total DNA and ATP, ALP and osteocalcin quantification. As for, has gene expression been reported on the ddCT or the 2^{-ddCt} ?

As mentioned earlier, we have included a statement to cover all this on the section “Cell culture in 3D printed scaffolds under sonic stimulation” that reads:

“All the experiments and analysis were performed in triplicates, n = 3.”

We have also included this information on each of the figure captions.

We have further specified the type of analysis on the section “Gene expression analysis” that reads:

“Ct values of RT-PCR were normalized against the housekeeping gene (GAPDH) and analyzed using the $2^{-(\Delta Ct)}$ model to show relative gene expression.”

As for the statistical analysis paragraph, it should be better described, specifying the type of distribution of the data and their variance, so that the assumptions for conducting the ANOVA test are not violated. Also, many of the items flagged in the 'reporting summary' for Statistics are not defined in the text at all. I deduced from the results the factors involved in the ANOVA analysis and they are not defined in the statistical paragraph. There are no covariates. For the multiple comparison test was a correction for the p-values adopted? What α level was chosen to define the type I error, $p < 0.1$, $p < 0.05$ or $p < 0.01$? If $p < 0.01$ remove the significance of $p < 0.1$.

We acknowledge the reviewer for bringing this to our attention. We have corrected the statistical analysis paragraph to provide further details.

"Statistical Analysis

Statistical analyses were performed using Graph Pad Prism 7.04. Sample size, significance and information about statistical tests are provided in the figure legends. Data are presented as mean \pm SD. For biological experiments (gene expression analysis, DNA and cell number measurements, ATP, ALP and osteocalcin release) data were analyzed by regular two-way ANOVA without repeated measures or matching followed by Tukey's post hoc corrected for multiple comparisons test. For particle analysis a one-way ANOVA without matching or pairing, followed by Tukey's post hoc corrected for multiple comparisons test was used. $\alpha = 0.05$ for all tests."

We have further modified the figure captions on the main text to include the sample size. On the SI, we have included in the figure captions the type of test, sample size and p values, when missing.

Figures

Pay attention to how the results are reported. Decimals must be indicated with the '.' and not with the ','.

We apologize for this mistake. We have corrected this mistake throughout the manuscript.

Figures 3S and 4S. Better adapt the characters of the text of the axes (for the x axis write 20:80,80,...). Note that in addition to the median, all data are reported as a beeswarm plot. I don't understand how 95% CI is displayed when not seeing specific error bars.

We appreciate the careful review. We have modified the axes of figure S3 and S4. The bars corresponding to the 95% CI are displayed on the figure but they are small and might not be visible by naked eye.

Figure S3. Analysis of the area (a) and aspect ratio (b) of the phase segregated particles on the cross-sectional direction for PLA:PCL ratios of 20:80, 30:70, 40:60, 60:40, 70:30 and 80:20. Bars show median and 95% CI. Statistical significance was calculated from 1-way ANOVA. (****) $p < 0.0001$, (***) $p < 0.001$, (**) $p < 0.01$ and (*) $p < 0.05$. $n = 5$.

*"Figure S4. Analysis of the area (a) and aspect ratio (b) of the phase segregated particles on the longitudinal direction of the scaffold fibers for PLA:PCL ratios of 20:80, 30:70, 40:60, 60:40, 70:30 and 80:20. Bars show median and 95% CI. Statistical significance was calculated from 1-way ANOVA. (****) $p < 0.0001$, (***) $p < 0.001$, (**) $p < 0.01$ and (*) $p < 0.1$. $n = 5$ "*

When reporting the data with $p <$, the values 0.0005, 0.005, 0.05 or 0.0001, 0.001, 0.01 are used by convention; it doesn't make much sense to write $p < 0.002$ or $p < 0.0021$ etc. because then we write directly $p = 0.0021$. In S3 and S4 the *** $p < 0.0002$ is not in the graphs. Also there must be an error in the values of p which are equal for the two figures.

Figures 2, S12 and S13 are missing the specifications of the asterisks values.

We thank the reviewer for the careful evaluation of the manuscript and apologize for the mistake. Indeed, there was an error that is now corrected. We have standardized to (****) $p < 0.0001$, (***) $p < 0.001$, (**) $p < 0.01$ and (*) $p < 0.1$ and included details on the SI figures that were missing them.

Tables.

In table 1 being a list of human genes, in the first column the genes should be written in uppercase and italics

We apologize for this mistake. We have amended the table showing the genes in uppercase and italics.

Table S1 and S2 do not report the number of tests and for S1 and S3 it is not clear whether they are means without standard deviations or single data.

We thank the reviewer for the careful analysis of the manuscript. We have provided these details on the corresponding figure captions as follows.

"Table S1. Measured frequency and amplitude of the different generated ultrasound waves. Set-ups 2, 6 and 10 were used for cell work and are referred as 10, 20 and 40 kHz. Data corresponds to the representative FFTs shown in Figure S7."

"Table S2. Flexural modulus (MPa) calculated for the different scaffolds compositions; PLA, PCL and Janus. Data is shown as mean \$\pm\$ SD, \$n = 5\$."

"Table S3. Measured frequencies and intensities of the ultrasound waves transmitted by the scaffolds. Data corresponds to the representative FFTs shown in Figure S10."

Reviewers' Comments:

Reviewer #3:

Remarks to the Author:

Accepted in the present form

Reviewer #4:

Remarks to the Author:

Overall I agree with the previous reviewers that the topic of the manuscript is interesting- the potential for active or acoustic-driven simulation of cells is an emerging area that shows great potential and is likely to be of interest to a broad audience. In general, the paper has been well constructed and puts forward a logical study from development and characterisation of the material system, to basic MSC characterisation and finally some interesting insights towards one of the mechanisms through which the cellular response is regulated.

I would like to have seen more discussion on the rationale for the specific stimulation used for the cells expts (30min/day)- although the authors state that constant stimulation would likely stress the cells, there is no real reason given for the specific choice of 30 min/day. Of course the combinations that could be tested are almost infinite, but why these particular conditions instead, for example, of 15 min or 1-2 hrs/day or 30 min 2x day?

Evaluation of response to reviewer 1:

1.

(1) Phase separation of PCL/PLA- this has been adequately addressed.

(2) It is evident that it is impractical to analyse the 50:50 composition by the same means as the other blends. It does seem odd to ignore this in the figures and so perhaps would be better to include the 50:50 and denote that the sample is not included in this analysis. The new analysis is helpful in addressing this point.

(3) Ultrasound transducers are not my area of expertise so I can't comment on the validity of the calculations, but the added explanation is helpful for understanding of the system in general.

(4) Can staining of the fibronectin-coated scaffolds be shown to confirm that the coverage is at least broadly equivalent over both of the polymers in the Janus strands?

2. The quantitative aspect of the data seems sufficient and I do not find that the study is overly reliant on qualitative assessments.

(1) S12 addresses this point sufficiently.

(3) Actually I disagree with the first reviewer. MTT data will give an average based on the cell population and requires the assumption of equal metabolic activity of the cells under all conditions. There are indications that acoustic stimulation changes the metabolic rate of cells independent of their viability (<https://onlinelibrary.wiley.com/doi/full/10.1002/adv.201902326>) and so direct counting is likely the more accurate method to determine cell numbers in this instance.

3. The comment on the different morphologies of the HAP is ok but there is quite a large leap to the composition that follows from this observation. Without direct analysis that the crystals are hydroxyapatite rather than Ca, this needs to be carefully stated as a possibility rather than fact. The current wording skates on the very edge of this but could likely benefit from some slight rephrasing to make it more evident this is at present "intelligent speculation".

4. Has been adequately addressed.

5. The potential for translation is an interesting and critical question- however, it is outside the scope of this paper and the discussion on this point is sufficient.

6. Decoupling the material from the mechanical stimulation would be the ideal but I agree with the

authors that there is no sensible way to do this. The comparison of the different materials in non-stimulated vs stimulated is sufficient in my view.

7. Has been adequately addressed.

Reviewer #3

Accepted in the present form

We acknowledge the reviewer for the positive evaluation of our manuscript.

Reviewer #4

Overall I agree with the previous reviewers that the topic of the manuscript is interesting- the potential for active or acoustic-driven simulation of cells is an emerging area that shows great potential and is likely to be of interest to a broad audience. In general, the paper has been well constructed and puts forward a logical study from development and characterisation of the material system, to basic MSC characterisation and finally some interesting insights towards one of the mechanisms through which the cellular response is regulated.

We acknowledge the reviewer for the overall positive evaluation of our manuscript.

I would like to have seen more discussion on the rationale for the specific stimulation used for the cells expts (30min/day)- although the authors state that constant stimulation would likely stress the cells, there is no real reason given for the specific choice of 30 min/day. Of course the combinations that could be tested are almost infinite, but why these particular conditions instead, for example, of 15 min or 1-2 hrs/day or 30 min 2x day?

We acknowledge the reviewer for the careful evaluation of the manuscript. There is yet not consensus by the scientific community on the optimal ultrasound parameters, neither the duration of the stimulation. We therefore choose these ultrasound stimulation duration based on previous in vitro studies that use 20 or 30 minutes of stimulation daily. We have included this on the manuscript that now reads:

Page 6, line 1: *"We seeded hBMSCs on the different scaffolds and cultured them for 7 days applying 30 minutes of stimulation daily at either 0, 10, 20 or 40 kHz using the same structures as modelled in figure 2b (Figure 2c, d, e and Figure S11), following the duration of previously reported ultrasound stimulation protocols^{19, 21}."*

Evaluation of response to reviewer 1:

1.

(1) Phase separation of PCL/PLA- this has been adequately addressed.

We acknowledge the reviewer for the positive assessment of our response.

(2) It is evident that it is impractical to analyse the 50:50 composition by the same means as the other blends. It does seem odd to ignore this in the figures and so perhaps would be better to include the 50:50 and denote that the sample is not included in this analysis. The new analysis is helpful in addressing this point.

We acknowledge the reviewer for bringing this point to us. We have modify the figure to include the 50:50 condition and denoted on the legend that this samples was not analysed. We have corrected this in SI Figures 3 and 4.

“Figure S3. Analysis of the area (a) and aspect ratio (b) of the phase segregated particles on the cross-sectional direction for PLA:PCL ratios of 20:80, 30:70, 40:60, 60:40, 70:30 and 80:20. Sample 50:50 is not included on the analysis. Bars show median and 95% CI. Statistical significance was calculated from 1-way ANOVA. (****) $p < 0.0001$, (***) $p < 0.001$, (**) $p < 0.01$ and (*) $p < 0.1$. $n = 5$.”

“Figure S4. Analysis of the area (a) and aspect ratio (b) of the phase segregated particles on the longitudinal direction of the scaffold fibers for PLA:PCL ratios of 20:80, 30:70, 40:60, 60:40, 70:30 and 80:20. Sample 50:50 is not included on the analysis. Bars show median and 95% CI. Statistical significance was calculated from 1-way ANOVA. (****) $p < 0.0001$, (***) $p < 0.001$, (**) $p < 0.01$ and (*) $p < 0.1$. $n = 5$ ”

(3) Ultrasound transducers are not my area of expertise so I can't comment on the validity of the calculations, but the added explanation is helpful for understanding of the system in general.

We acknowledge the reviewer for the positive assessment of our response.

(4) Can staining of the fibronectin-coated scaffolds be shown to confirm that the coverage is at least broadly equivalent over both of the polymers in the Janus strands?

The scaffolds for cell culture experiments were coated with vitronectin. While this experiment would indeed be possible, we have included the PCL and PLA scaffold experiments to overcome any issue associated to chemical differences on the scaffolds surface. Fluorescent microscopy images show homogeneous cell coverage over the scaffolds surface in all three sample types. Moreover, all the data shown was normalized for the DNA content and data related to proliferation was normalized to the number of cells at 0 MHz stimulation. Thus, we believe that coating would not play a significant role in our experimental set-up. No further changes have been done to the manuscript.

2. The quantitative aspect of the data seems sufficient and I do not find that the study is overly reliant on qualitative assessments.

We acknowledge the reviewer for the positive evaluation of the manuscript.

(1) S12 addresses this point sufficiently.

We acknowledge the reviewer for the positive assessment of our response.

(3) Actually I disagree with the first reviewer. MTT data will give an average based on the cell population and requires the assumption of equal metabolic activity of the cells under all conditions. There are indications that acoustic stimulation changes the metabolic rate of cells independent of their viability (<https://onlinelibrary.wiley.com/doi/full/10.1002/advs.201902326>) and so direct counting is likely the more accurate method to determine cell numbers in this instance. We agree with the reviewer and acknowledge the positive assessment of our response.

3. The comment on the different morphologies of the HAP is ok but there is quite a large leap to the composition that follows from this observation. Without direct analysis that the crystals are hydroxyapatite rather than Ca, this needs to be carefully stated as a possibility rather than fact. The current wording skates on the very edge of this but could likely benefit from some slight rephrasing to make it more evident this is at present "intelligent speculation".

We agree with the reviewer and have toned down some sentences that now read:

Page 9, line 3: *"However, a detailed observation of the deposited minerals by SEM revealed that hBMSCs cultured on PCL underwent what has been proposed as "chemical differentiation", depositing minerals with a smooth and big crystal-like morphology characteristic of Ca²⁺ salts³⁵. Minerals deposited on Janus scaffolds, however, showed a rounded and porous morphology typically ascribed to amorphous hydroxyapatite or calcium phosphate^{35, 36}. During bone mineralization, amorphous and globular shaped calcium phosphates undergo mineralization to form carbonated hydroxyapatite, thus suggesting the bone formation potential on Janus scaffolds^{37, 38, 39, 40}."*

4. Has been adequately addressed.

We acknowledge the reviewer for the positive assessment of our response.

5. The potential for translation is an interesting and critical question- however, it is outside the scope of this paper and the discussion on this point is sufficient.

We acknowledge the reviewer for the positive assessment of our response.

6. Decoupling the material from the mechanical stimulation would be the ideal but I agree with the authors that there is no sensible way to do this. The comparison of the different materials in non-stimulated vs stimulated is sufficient in my view.

We acknowledge the reviewer for the positive assessment of our response.

7. Has been adequately addressed.

We acknowledge the reviewer for the positive assessment of our response.